# Distinct roles for *S. cerevisiae* H2A copies in recombination and repeat stability, with a role for H2A.1 threonine 126

Nealia CM House[1†‡], Erica J Polleys[1†], Ishtiaque Quasem[1], Marjorie De la Rosa Mejia[1], Cailin E Joyce[1§], Oliver Takacsi-Nagy[1#], Jocelyn E Krebs[2], Stephen M Fuchs[1], Catherine H Freudenreich[1,3]*

[1]Department of Biology, Tufts University, Medford, United States; [2]Department of Biological Sciences, University of Alaska Anchorage, Anchorage, United States; [3]Program in Genetics, Graduate School of Biomedical Sciences, Tufts University, Boston, United States

*For correspondence:
catherine.freudenreich@tufts.edu

[†]These authors contributed equally to this work

Present address: [‡]Department of Radiation Oncology, Division of Genomic Stability, Dana-Farber Cancer Institute, Harvard Medical School, Boston, United States; [§]Agenus Incorporated, Lexington, United States; [#]ArsenalBio, San Francisco, United States

**Abstract** CAG/CTG trinucleotide repeats are fragile sequences that when expanded form DNA secondary structures and cause human disease. We evaluated CAG/CTG repeat stability and repair outcomes in histone H2 mutants in *S. cerevisiae*. Although the two copies of H2A are nearly identical in amino acid sequence, CAG repeat stability depends on H2A copy 1 (H2A.1) but not copy 2 (H2A.2). H2A.1 promotes high-fidelity homologous recombination, sister chromatid recombination (SCR), and break-induced replication whereas H2A.2 does not share these functions. Both decreased SCR and the increase in CAG expansions were due to the unique Thr126 residue in H2A.1 and *hta1Δ* or *hta1-T126A* mutants were epistatic to deletion of the Polδ subunit Pol32, suggesting a role for H2A.1 in D-loop extension. We conclude that H2A.1 plays a greater repair-specific role compared to H2A.2 and may be a first step towards evolution of a repair-specific function for H2AX compared to H2A in mammalian cells.

## Introduction

Integral to the eukaryotic DNA damage response is chromatin structure modifications surrounding the break site (reviewed in *House et al., 2014a*; *Seeber et al., 2013*; *Price and D'Andrea, 2013*). In response to DNA double strand breaks (DSBs) and stalled or collapsed replication forks, the SQEL motif in the H2A C-terminal tail is phosphorylated at Ser129 (H2AX-Ser139 in mammals) by the Phosphoinositide 3-Kinase-Related Kinases (PIKKs), Mec1 and Tel1 (ATR and ATM in mammals) (*Downs et al., 2000*; *Lisby et al., 2004*). This modification, termed γH2A (γH2AX in mammals), marks the site of damage and is propagated along the chromatin, detectable up to 50 kb from the break site in yeast (*Shroff et al., 2004*) and megabases in mammalian cells (*Rogakou et al., 1999*).

γH2A/γH2AX occurs rapidly upon induction of a DSB, detectable within minutes after damage (*Shroff et al., 2004*; *Rogakou et al., 1999*; *Paull et al., 2000*), and is required to initiate the cascade of histone modifications, chromatin remodeling, and repair factor recruitment and retention necessary for repair. However, other modifiable residues in the H2A tails can also modulate repair factor binding: H2A/H2AX K5ac, K15ub/ac, K36ac, K119ub, and Y142ph all contribute to DNA repair (reviewed in *Hunt et al., 2013*; *Jacquet et al., 2016*).

Whereas in humans over a dozen H2A variants have been identified (*Albig et al., 1999*; *Bönisch and Hake, 2012*), *S. cerevisiae* contains just three variants of H2A, encoded by *HTA1*, *HTA2*, and *HTZ1*. *HTA1* and *HTA2* encode canonical H2A and the two copies are nearly identical in amino acid sequence except for a direct alanine-threonine switch at positions 125/126 in the C-terminal tail (Figure 1B); the underlying DNA sequence is 94% similar. H2A-T126 is phosphorylatable in

vivo, even in the absence of DNA damage (*Wyatt et al., 2003*; *Moore et al., 2007*). The third H2A variant, H2A.Z, has only 56% amino acid sequence homology to canonical H2A.

H2A modification is a major contributor to DNA repair and may be particularly important in promoting efficient repair at unstable genomic elements. CAG/CTG trinucleotide repeats are in this category, as they can form abnormal secondary structures, such as hairpins and slip-stranded DNA (reviewed in *McMurray, 1999*; *Usdin et al., 2015*; *Schmidt and Pearson, 2016*), and break at a higher frequency than non-repetitive DNA (*Freudenreich et al., 1998*; *Callahan et al., 2003*; *Nasar et al., 2000*). Repair or replication errors within the CAG/CTG repeat can lead to instability, or a change in repeat units. Once expanded (addition of repeat units), the repeat tract is increasingly unstable and prone to further expansion in a length-dependent manner (reviewed in *Usdin et al., 2015*; *Kim and Mirkin, 2013*). CAG/CTG repeats are found throughout the human genome but repeat expansion beyond a threshold length of approximately 35 repeats can lead to human disease, including Huntington's disease, myotonic muscular dystrophy, and several spinocerebellar ataxias (*Usdin et al., 2015*; *Mirkin, 2007*). The CAG/CTG repeat is a strong nucleosome-positioning element, shown in vitro by nucleosome assembly assays and visualized by electron microscopy (*Godde and Wolffe, 1996*; *Wang et al., 1994*). The intrinsic nucleosome-positioning characteristic of the CAG repeat makes this an interesting and sensitive sequence at which to study the chromatin environment during DNA repair. Further, the unstable nature of the repeat allows us to experimentally test the importance of chromatin and repair factors in promoting high-fidelity repair, since repair errors (errors in synthesis, alignment, processing, etc) can lead to repeat tract length changes.

Secondary structures that occur at CAG/CTG repeats can interfere with DNA transactions, causing stalled or collapsed replication forks, gaps, nicks, and DSBs (*Usdin et al., 2015*). Repair can proceed via homologous recombination (HR), but this repair itself can be a source of mutagenesis if it does not proceed with high fidelity (reviewed in *Polleys et al., 2017*). Several steps during HR presumably require nucleosome repositioning or eviction, including resection, strand invasion, copying the template and D-loop extension, and resetting the chromatin structure after repair. Efficient completion of each stage of HR is expected to be important to prevent errors that lead to CAG repeat expansions (*Polleys et al., 2017*).

We previously described a role for histone H4 acetylation in promoting high-fidelity HR during post-replication repair at CAG repeats (*House et al., 2014b*). Here, we explore the role of histone H2A in CAG repeat maintenance. In a primary genetic screen for CAG repeat fragility and a secondary screen of CAG repeat instability, deletion of histone H2A.1 increased CAG repeat fragility and expansion frequency. However, deletion of the second copy of this protein, H2A.2, had no effect on repeat fragility or instability. Since histone H2A could be participating in one or more pathways that contribute to repeat stability, several hypotheses have been explored to explain this discrepancy. We found that H2A.1 threonine 126 (T126) is required to prevent CAG expansions and that expansions that arise in the absence of phosphorylatable T126 are dependent on Rad51, Rad52, Rad57, and the Polδ subunit, Pol32. In addition, we show that H2A.1 and H2A.1-T126 are required for efficient SCR at non-repetitive DNA sequences, and are working in the same pathway as the Polδ subunit Pol32. Finally, we show that H2A.1 is specifically important in mediating repair via BIR. Together, these results demonstrate that H2A.1 plays an important role in promoting efficiency and fidelity of recombination during repair. This role is distinct from H2A.2, and our results implicate the T126 residue as important for this distinction.

## Results

### H2A histone variants contribute differentially to $(CAG)_{85}$ repeat stability

To identify important factors for maintaining expanded CAG/CTG trinucleotide repeats, a screen was performed for genes that protect against repeat fragility using a yeast artificial chromosome (YAC) end loss assay in the Matα haploid deletion set (screen originally described in *Gellon et al., 2011*; assay illustrated in *Figure 1—figure supplement 1A* and reviewed in *Polleys and Freudenreich, 2018*; *Polleys and Freudenreich, 2020*). Initial semi-quantitative and quantitative assays showed a 2-fold increase in the rate of 5-FOA-resistance (FOA$^R$) in the *hta1Δ* mutant compared to the wild-type for a strain containing a YAC with a $(CAG)_{85}$ repeat tract, whereas the *hta2Δ* mutant

did not deviate from wild-type. Upon further analysis using multiple independent *hta1Δ* transformants, we observed a wide range of repeat fragility rates and thus were unable to statistically verify the increase over wild-type (*Figure 1C*; *Supplementary file 1*). The *hta1Δ* mutant is mildly sensitive to phleomycin and not sensitive to the other DNA damaging agents tested (MMS, CPT, HU)

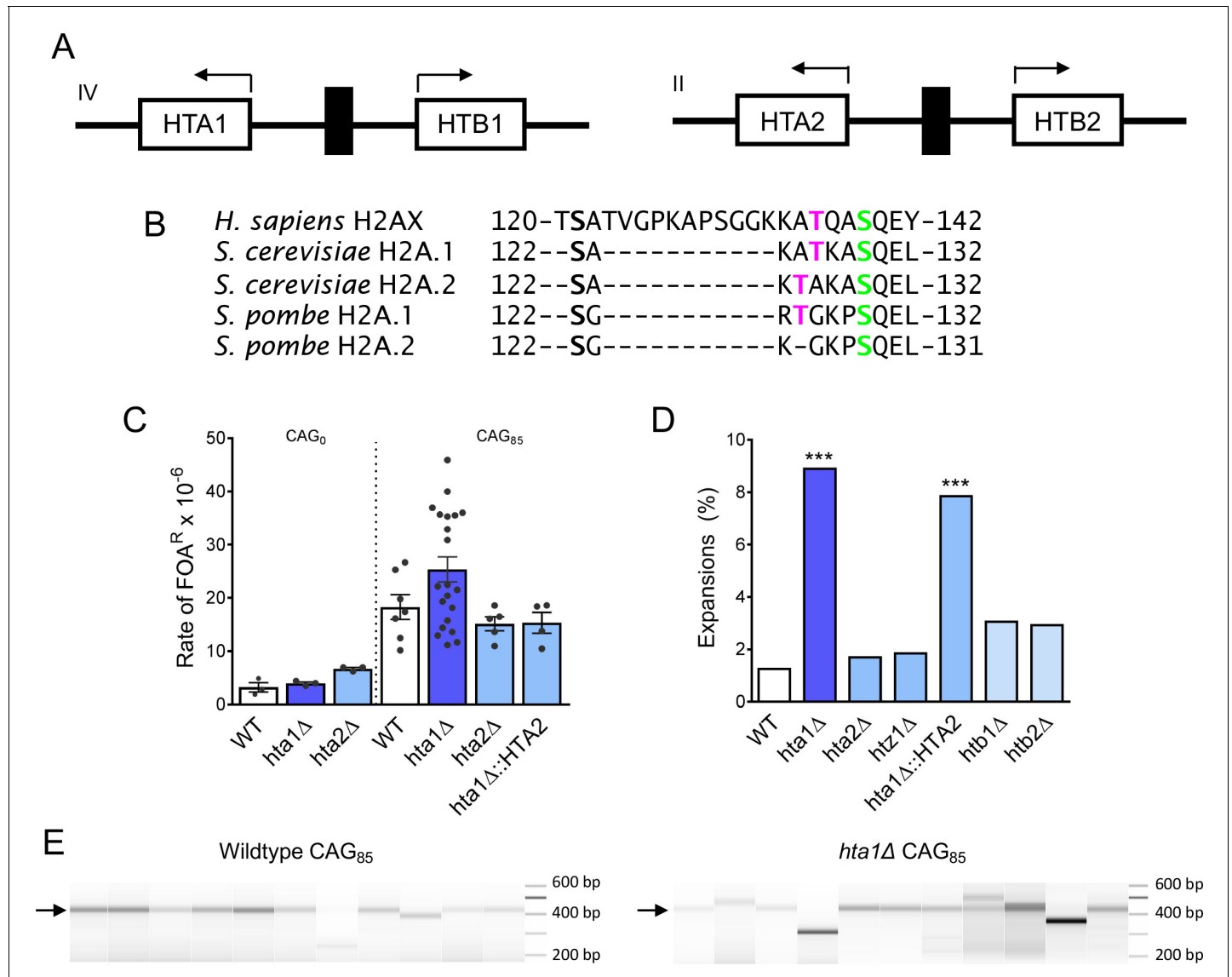

**Figure 1.** H2A.1 is required to prevent (CAG)$_{85}$ repeat expansions. (A) *HTA* is present in two copies in *S. cerevisiae*, paired with *HTB*. The gene pairs are divergently transcribed. The *HTA1-HTB1* locus is on chromosome IV, the *HTA2-HTB2* locus is on chromosome II. (B) H2A/H2AX protein sequences were aligned by Clustal Omega, only the C-terminal tail residues (past the histone fold) are shown. The threonine residue at position 125/126 is present in human H2AX, conserved in *S. cerevisiae* and *S. pombe* H2A.1. (C) Fragility of YAC CF1 (either contained (CAG)$_0$ (no tract) or a (CAG)$_{85}$ repeat tract) was assessed as previously described (reviewed in *Polleys and Freudenreich, 2018*) (*Figure 1—figure supplement 1A*). Rates were evaluated for significant differences by the Student's t-test; no differences were statistically significant; error bars represent SEM (*Supplementary file 1*). (D, E) CAG repeat expansion (D) and contraction (E) frequencies were measured by a PCR-based stability assay, using primers P1 and P2, described in *Figure 1—figure supplement 1A*. Products were run in high-resolution Metaphor agarose or a fragment analyzer (as shown in F) and repeat length changes were evaluated. Raw instability data can be found in *Supplementary file 2*. Statistical deviation from wild-type was calculated by Fisher's Exact Test: ***p<0.001 F) Visualization of CAG repeat lengths using an Advanced Analytical fragment analyzer. Arrow indicates length of (CAG)$_{85}$ PCR product. The online version of this article includes the following figure supplement(s) for figure 1:

**Figure supplement 1.** YAC system to evaluate CAG repeat fragility and instability.

**Figure supplement 2.** Drug sensitivity panels in H2A mutant strains.

(*Figure 1—figure supplement 2*). We conclude that there is likely a mild defect in repair at the CAG repeat in the absence of H2A copy 1.

We next evaluated the contribution of all the histone H2 variants to CAG/CTG repeat stability. Histone genes were deleted and $(CAG)_{85}$ repeat tract length changes were monitored by PCR analysis (*Callahan et al., 2003*; *House et al., 2014b*) (*Figure 1D,E,F*, *Supplementary file 2*). Deletion of the genes encoding the two copies of H2A differentially affect CAG repeat stability: expansion frequency is significantly increased in the *hta1Δ* mutant (7-fold increase over wild-type, $p=2.7\times10^{-5}$) while expansion frequency in the *hta2Δ* mutant is not significantly changed from wild-type (1.3-fold increase over wild-type, p=0.71) (*Figure 1D*). Contractions were increased 2.6-fold in the *hta1Δ* but not the *hta2Δ* mutant (*Figure 1E*). Thus, H2A.1 is required to suppress CAG instability while H2A.2 is not.

The H2A.Z variant is encoded by the gene *HTZ1* in yeast. Since H2A.Z is deposited at a DSB during repair (*Kalocsay et al., 2009*), we tested whether it is required to maintain CAG repeat stability. Although the *htz1Δ* mutant is sensitive to DNA damaging agents (*Morillo-Huesca et al., 2010*; *Papamichos-Chronakis et al., 2011*), deleting the *HTZ1* gene did not affect repeat expansion frequency, though contractions were significantly increased in the *htz1Δ* mutant (2.1-fold over WT, $p=3.1\times10^{-6}$) (*Figure 1E*) (*House et al., 2014b*). Deleting either *HTB* gene had no significant effect on repeat expansion frequency (*Figure 1D*), but contractions were somewhat elevated in the *htb1Δ* mutant (2-fold over WT, p=0.01)(*Figure 1E*). Therefore, H2A.1, H2A.Z, and H2B.1 are required to prevent CAG contractions. However, of the histone H2 proteins, only H2A.1 plays a significant role in preventing CAG repeat expansions, suggesting some specialized role for H2A.1 that we have explored further.

## H2A.1 sequence, not histone levels, confers specificity to its role in preventing CAG repeat expansions

Histone H2A is encoded by two nearly identical gene copies in *S. cerevisiae,* each paired with a copy of H2B and differentially regulated (*Norris and Osley, 1987*) (*Figure 1A*). The H2A.1 and H2A.2 protein sequences are identical except for a direct threonine/alanine (T/A) switch in the C-terminal tail (*Figure 1B*). While transcription from the *HTA1-HTB1* gene pair can be upregulated in the absence of *HTA2*, *HTA2-HTB2* is transcribed at a constant rate (*Moran et al., 1990*). As a result, the H2A pools will be normal in an *hta2Δ* mutant, whereas an *hta1Δ* mutant may have a global decrease in H2A. A second pathway of gene dosage compensation exists in which the *HTA2-HTB2* gene pair amplifies to form a minichromosome (that also contains the *HHT1-HHF1* (H3-H4) gene pair) in the absence of *HTA1-HTB1* (35). To distinguish if repeat stability is mediated by the H2A protein sequence or histone levels, the *HTA2* sequence was placed under the control of the *HTA1* promoter, replacing the *HTA1* gene (*hta1Δ::HTA2)*. In this strain, the H2A.2 protein will be expressed at the same level and timing as H2A.1 in a wild-type cell but the H2A.1 protein will not be present. Equal expression of H2A proteins was confirmed by Western blot (*Figure 1—figure supplement 1B*). We tested strains for gene amplification of *HTA1* or *HTA2* by qPCR and no instances of gene amplification were detected at the genomic level (*Supplementary file 8*). If H2A expression level or timing is the major contributor to repair of the CAG repeat, repeat maintenance will be at wild-type levels when *HTA2* is expressed from the *HTA1* promoter. However, we observed that CAG repeat expansions remained significantly increased from the wild-type in the *hta1Δ::HTA2* strain ($p=4.2\times10^{-4}$ to WT, *Figure 1D*), though a partial suppression of contractions was observed (*Figure 1E*) (*Supplementary file 2*). Therefore, while CAG contractions appear to be sensitive to overall histone levels, H2A.2 cannot compensate for H2A.1 in preventing repeat expansions, even when expressed at H2A.1 levels under control of the *HTA1* promoter. We conclude that the sequence of H2A.1, not histone levels or subtleties in expression timing, is required to prevent CAG repeat expansions.

## Nucleosome positioning at the CAG repeat is maintained in H2A mutants

Since H2A.2 is not upregulated in the absence of H2A.1, we hypothesized that an *hta1Δ* mutant may cause local histone depletion or disruption of the chromatin structure at the CAG repeat tract, leading to instability. Although overall bulk chromatin structure was not altered in an *hta1Δhtb1Δ* strain, some areas of the genome were more sensitive to micrococcal nuclease (MNase) digestion

(*Norris et al., 1988*). The CAG repeat is a strong nucleosome positioning element and thus could be more sensitive to H2A.1 depletion than other regions of the genome. To visualize nucleosome positioning at the CAG repeat, we used indirect end-labeling with a probe upstream of the CAG repeat (*Figure 1—figure supplement 1A*, red line) and measured sensitivity of the chromatin to MNase digestion. A series of discrete, protected fragments (~5 nucleosomes), is observed, indicating several positioned nucleosomes within and flanking the CAG repeat (*Figure 2A*). The pattern is consistent with canonical ~165 bp spacing between nucleosomes, and suggests there is no visible, major disruption to the chromatin structure in the *hta1Δ* cells compared to the wild-type (*Figure 2A*).

To generate a high-resolution nucleosome map of the regions flanking the CAG repeat, we used a custom Illumina BeadArray containing probes spanning 425 bp upstream to 438 bp downstream of the CAG repeat tract on the YAC, including two CAG repeat containing probes and one pure CAG repeat probe, in non-overlapping 30-mers (*Supplementary file 3*). We hybridized MNase digested mononucleosomes of *hta1Δ*, *hta2Δ*, and wild-type (CAG)$_{85}$ cells to the array to measure nucleosome protection at the CAG repeat. The CAG repeat-containing probes produced a peak in intensity compared to the flanking, non-CAG repeat-containing probes, indicating strong nucleosomal protection at the CAG repeat (*Figure 2B*). This result confirms previous in vitro data that the CAG repeat is a strong nucleosome binding sequence (*Wang et al., 1994*; *Wang and Griffith, 1995*; *Volle and Delaney, 2012*) and that it positions a nucleosome in vivo on a yeast chromosome. The protection was not reduced in the *hta1Δ* and *hta2Δ* mutants (*Figure 2B*). Thus, both methods used show that there is a positioned nucleosome at the CAG tract that is not altered in the *hta1Δ* background. We conclude that nucleosome positioning is not the major contributing factor to CAG instability in the *hta1Δ* mutant; however, subtle differences in chromatin structure in the absence of H2A.1 that are not visible by these assays may still impact repair pathway selection.

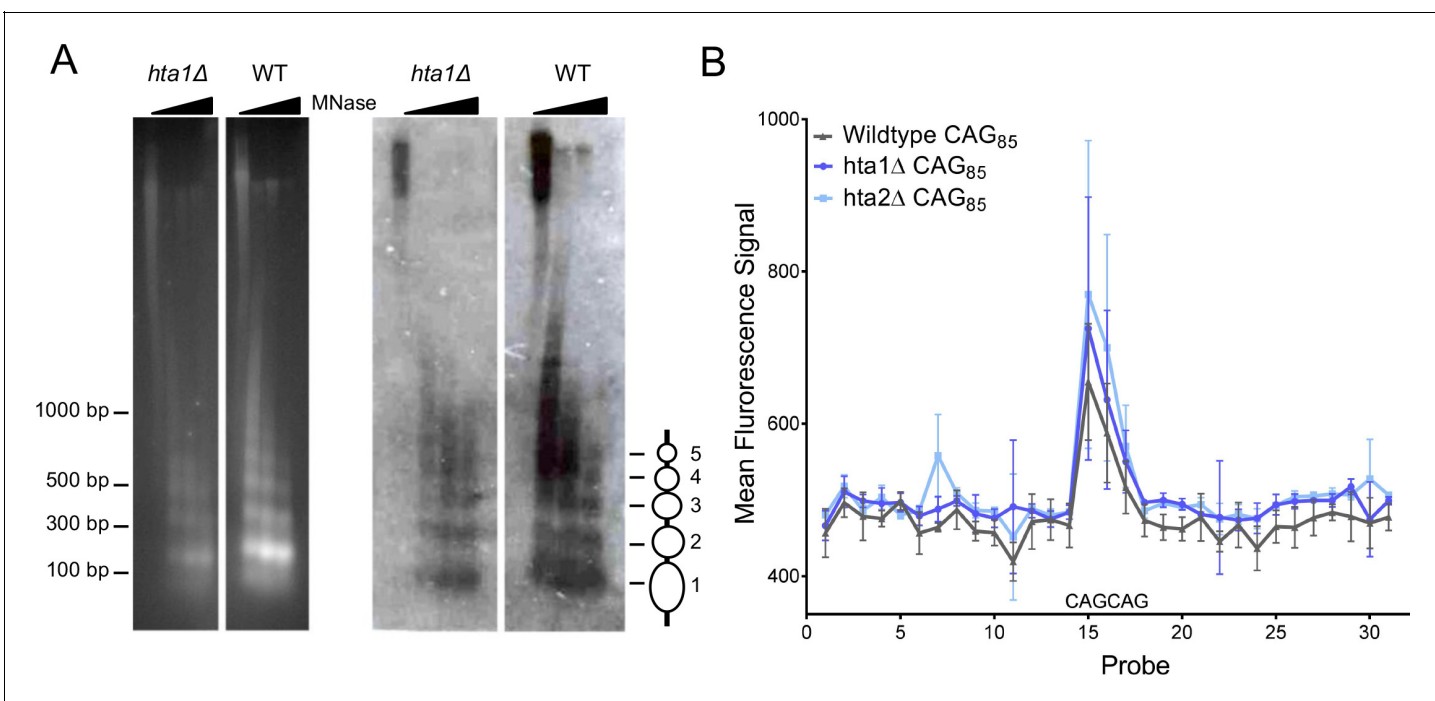

**Figure 2.** Nucleosome positioning at a (CAG)$_{85}$ repeat is not altered in the absence of H2A.1 or H2A.2. **A)** Indirect end-labeling of nucleosomal DNA upstream of the CAG repeat. MNase (0, 0.25, 2.5, and 7.5 units) digested DNA was run in 1.5% agarose with ethidium bromide (left) and Southern blotted (right) using a probe ~100 bp proximal to the CAG repeat (red line *Figure 1—figure supplement 1A*). Ovals represent nucleosome positions. The experiment was repeated six times; a representative blot is shown. (**B**) Illumina array mapping of nucleosome protection at the CAG repeat. Mononucleosomal DNA from strains containing the (CAG)$_{85}$ repeats was hybridized to a custom array of 30-mer probes spanning 425 bp upstream of the repeat to 436 bp downstream of the repeat in YAC CF1. Probes 14–16 contain CAG repeats; probe 15 is composed purely of CAG repeats (probe sequences in *Supplementary file 3*). Error bars represent standard deviation of 2–3 independent experiments.

## The Thr126 residue in the H2A.1 C-terminal tail is required for $(CAG)_{85}$ repeat maintenance

H2A.1 and H2A.2 vary in amino acid sequence only by the position of threonine in the C-terminal tail, which occurs at either position 126 in H2A.1 or 125 in H2A.2 (*Figure 1B*). Previous work measuring steady state levels of $^{32}$P-labeled histones showed H2A phosphorylation was decreased ~2 fold in an *hta1-T126A* mutant, identifying T126 as a phosphorylatable residue (*Wyatt et al., 2003*). To address the role of specific residues in the H2A C-terminal tail in CAG stability, we introduced point mutations in the endogenous copy of *HTA1*. When T126 was rendered non-modifiable by mutation to alanine, expansions were significantly increased over wild-type (3.7-fold over WT, p=0.04 to WT) (*Figure 3A*; *Supplementary file 2*). Deletion of *HTA2* (*hta1-T126A hta2Δ*) had no further impact on CAG expansion frequency (*Supplementary file 2*). Expression of *hta1-T126A* from a plasmid in an *hta1Δhta2Δ* background resulted in a similar increase in expansions (3.1-fold over WT; p=0.04 to WT)(*Figure 3A*, right). Therefore, the H2A.1-T126 residue is required for efficient repair of the CAG repeat to prevent expansions. Since phosphorylation of T126 would be lost in the *T126A* mutant, this suggests the possibility that phosphorylation of this residue is important for its role in repair.

We next asked if constitutive phosphorylation of the T126 residue could promote high-fidelity repair to prevent CAG expansions by introducing the phospho-mimic *T126E* mutation into the endogenous *HTA1* gene. CAG expansion frequency remained elevated in the *T126E* mutant (4.9%, 3.7-fold over WT, p=0.01) (*Figure 3A*; *Supplementary file 2*). This result suggests that either dynamic phosphorylation of H2A.1-T126 or another characteristic of having a threonine at position 126 is important for promoting CAG stability. Given the specific role for H2A.1, this result suggests that the position of the threonine within the tail affects the efficiency with which the H2A copies contribute to DNA replication or repair. CAG contractions were relatively unaffected in both the *hta1-T126A* and *hta1-T126E* mutants (*Supplementary file 2*), therefore this residue primarily protects against expansions.

We also tested the importance of the H2A-S129 residue in repeat maintenance. Despite H2A-S129ph being preferentially detected at a $(CAG)_{155}$ tract in this same location by ChIP (*House et al., 2014b*), the ability to phosphorylate H2A-S129 did not significantly affect CAG repeat tract stability; neither expansion nor contraction frequencies were significantly altered from wild-type in the *hta1-S129A* mutant, expressed either from the genomic copy of *HTA1* or from a plasmid (*Figure 3A*, *Supplementary file 2*). The double *hta1-S129A/T126A* mutant expressed from the endogenous *HTA1* gene locus did not result in any further increase in expansion frequency from the *hta1-T126A* single mutant (*Figure 3A*; *Supplementary file 2*).

## H2A.1 is enriched at the CAG repeat during S-phase

We wanted to evaluate in vivo phosphorylation of the H2A-T126 residue and used a custom antibody raised against a phosphopeptide corresponding to the H2A C-terminal tail phosphorylated at T126. Although this antibody was specific to H2A-T126ph by peptide dot blot (*Figure 3—figure supplement 1A*) phosphatase treatment of cell extracts only resulted in a 30% reduction in antibody recognition (*Figure 3—figure supplement 1B*, *Supplementary file 9*). To test if this antibody could distinguish between H2A isoforms, recognition by the antibody was compared in wild-type (*HTA1* and *HTA2* both present), *hta1Δ* (only *HTA2* present), *hta2Δ* (only *HTA1* present) and *hta1-T126A* (genomic copy of *hta1* mutated, *HTA2* still present). The signal is significantly diminished in the *hta1Δ* and *hta1-T126A* strains to 20–30% of WT levels, but not in the *hta2Δ* mutant (*Figure 3B*. *Supplementary file 9*). We conclude that the antibody is specifically recognizing the H2A.1 protein isoform containing threonine at position 126 in the tail, with a low level of background reactivity to the H2A.2 isoform. Importantly, total H2A protein and *HTA* gene levels remain constant in each mutant (*Figure 1—figure supplement 1B*; *Supplementary file 8*) (*Libuda and Winston, 2006*). Since this antibody mainly detects H2A.1, but not H2A.2, we have designated it 'H2A.1$_{T126}$.'.

Since H2A.1 is specifically required to promote repair of the CAG repeats, we wanted to know if the H2A.1 histone variant can be detected at an expanded CAG repeat. We tested recruitment of H2A.1 to the CAG repeat by ChIP during S-phase at time points when we have previously seen measurable increases in repair factors (*House et al., 2014b*; *Sundararajan et al., 2010*). Cells containing a $(CAG)_{155}$ tract were α-factor arrested in $G_1$ and time points were taken after release into fresh media. Whereas H2A.1 levels remain relatively constant at an *ACT1* control locus, H2A.1 is

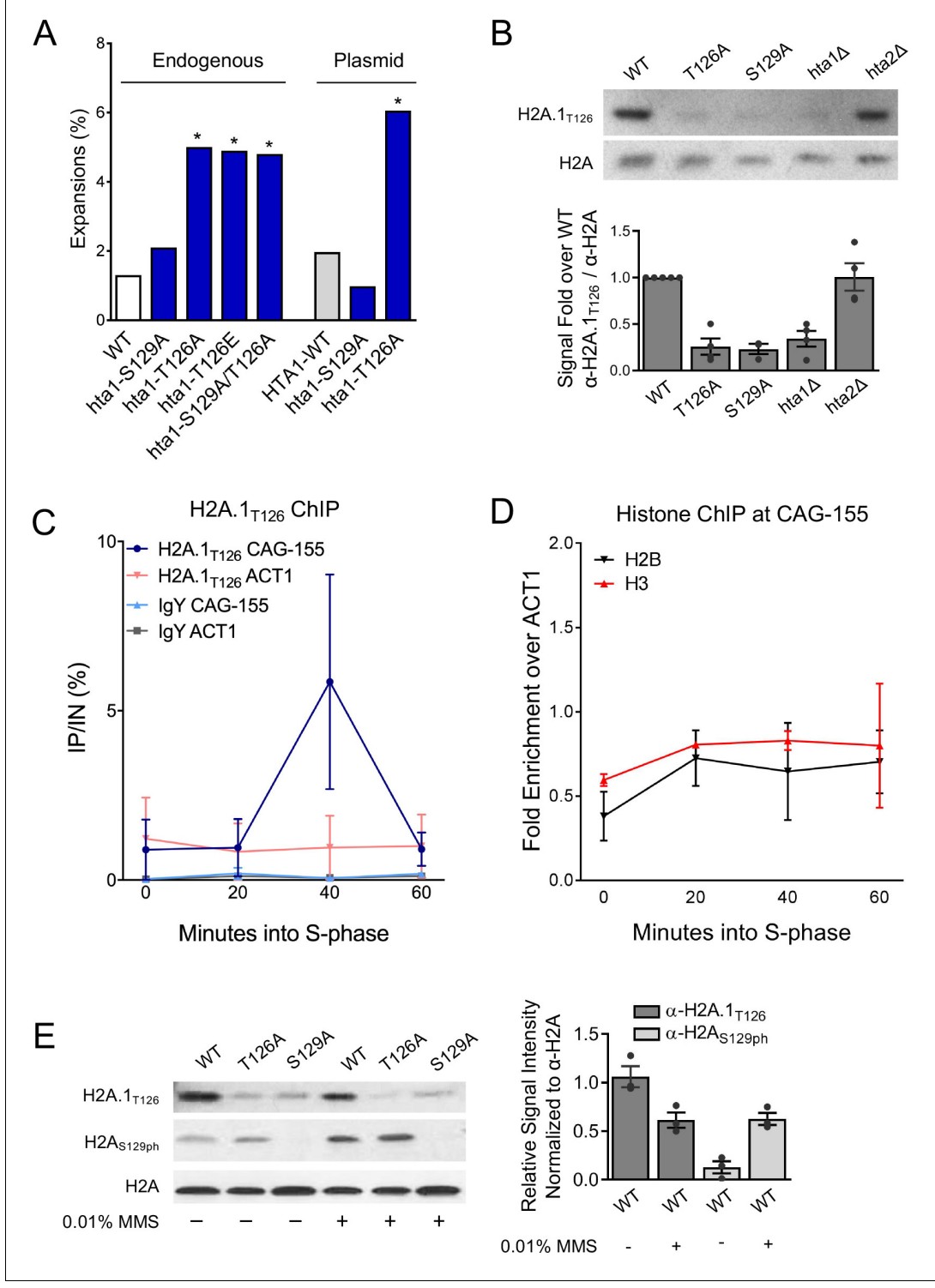

**Figure 3.** H2A.1-T126 is required to prevent $(CAG)_{85}$ repeat expansions and is phosphorylated in vivo. **A)** CAG repeat length changes were evaluated in strains with H2A.1 C-terminal tail point mutations. The source of H2A.1 is noted above the bars: either the point mutation was in the genomic copy of *hta1* (*HTA2* still present) or expressed from a plasmid (both endogenous copies of *HTA-HTB* are deleted). Statistical deviation from the wild-type was calculated by Fisher's Exact Test, *p<0.05. (**B**) A Western blot with protein prepared from the indicated strains was probed with antibody raised against the $H2A.1_{T126ph}$ C-terminal tail peptide ($H2A.1_{T126}$) or anti-H2A. H2A mutations were in the genomic copy of indicated H2A variant; the other variant was still present. Quantification of this blot and other experiments using the same strains (each indicated by a point) is shown at right. The average

*Figure 3 continued on next page*

*Figure 3 continued*

and SEM are plotted. (**C**) Incorporation of H2A.1 at a (CAG)$_{155}$ repeat was measured by ChIP using the H2A.1$_{T126}$ antibody; IgY was used as a negative control. Cultures were α-factor synchronized and released into fresh medium; time point taken as indicated after release, in S-phase. Enrichment was measured using a primers set that amplified a region spanning 0.4 and 0.6 kb upstream of the CAG repeat or at an *ACT1* internal control locus. The average IP/IN and standard error for two independent experiments is plotted. (**D**) ChIP for histones H2B and H3 at the CAG repeat was performed as in (**C**). IP/IN values were normalized to the *ACT1* control signal. Individual IP/IN and ACT1 normalized values can be found in **Supplementary file 4**. (**E**) Western blots using antibodies against H2A.1$_{T126}$ (top panel), H2A$_{S129ph}$ (middle panel), and total H2A (bottom panel). Total protein was prepared from H2A wild-type and point mutant strains (mutations in the plasmid copy of indicated H2A variant, *hta2Δ* background) after 2 hr exposure to either no drug (-) or 0.01% MMS (+). Quantification of the H2A.1$_{T126}$ and H2A$_{S129ph}$ antibody signals normalized to total H2A levels in WT strains are graphed to the right; dots indicate independent experimental replicates and the average and SEM is shown. Quantification of all bands is shown in and listed in **Supplementary file 9**.

The online version of this article includes the following figure supplement(s) for figure 3:

**Figure supplement 1.** Analyses of H2A.1$_{T126}$ antibody specificity and H2A.1 levels over the cell cycle.

**Figure supplement 2.** Analyses of H2A.1$_{T126}$ levels after exposure to DNA damaging agents and deletion of kinases involved in the DNA damage response.

specifically enriched at the CAG repeat 40 min into S-phase, when we previously saw peak γH2A at this location (*House et al., 2014b*), and returns to baseline by 60 min (*Figure 3C*). We also tested H2B and H3 recruitment to monitor the H2A-H2B dimer and overall nucleosome levels at the CAG repeat. We found no significant differences in H2B or H3 occupancy relative to an endogenous *ACT1* locus (*Figure 3D*), indicating that the recruitment of H2A.1 is specific and not an artifact of increased nucleosome occupancy at this site during S-phase. Western blot analysis of α-factor arrested and released cells showed no increase in H2A.1 across the cell cycle (*Figure 3—figure supplement 1C*). Interestingly, enrichment of H2A.1 at the CAG tract by ChIP is occurring 20 min after γH2A begins to accumulate and peak Mre11 enrichment at the CAG repeat is detected (*House et al., 2014b*; *Sundararajan and Freudenreich, 2011*). Therefore, incorporated H2A.1 peaks after initial damage signaling at the repeat and coincides with maximal levels of γH2A. This transient occupancy supports that H2A.1 is specifically incorporated during DNA repair.

To determine if H2A.1 expression is damage inducible, H2A.1 levels were monitored after exposure to MMS, a DNA base alkylating agent that causes abasic sites that can be converted into single and double strand breaks. In the presence of 0.01% MMS, H2A-S129ph is increased as expected, but H2A.1$_{T126}$ signal is in fact decreased and is therefore not induced by the DNA damage caused by low levels of MMS (*Figure 3E*). We also see no change in H2A.1$_{T126}$ signal when cells are treated with 0.2M HU + 0.03% MMS, a treatment that induces collapsed replication forks (*Nagai et al., 2008*) (*Figure 3—figure supplement 2A*). We conclude that H2A.1 expression is not induced by DNA damage, and tentatively conclude that H2A.1$_{T126}$ phosphorylation is also not likely not induced by DNA damage, since a 30% increase in antibody signal was not observed (as would be expected based on additional antibody recognition of phosphorylated form). A caveat to the conclusion of no damage inducibility is that S129ph could reduce the antibody recognition, precluding detection of an increased signal after MMS treatment. Nonetheless, the conclusion is supported by the lack of sensitivity of the H2A.1-T126A mutant to DNA damaging agents (*Figure 1—figure supplement 2*) and the fact that we were unable to detect a decrease in antibody signal when kinases known to phosphorylate targets in response to DNA damage and up-regulate the DNA damage response were mutated (*Figure 3—figure supplement 2C*).

We also tested if H2A-S129ph is altered in the *hta1-T126A* mutant. Importantly, recognition of H2A-S129ph is not impaired by the *hta1-T126A* mutation (*Figure 3E*). Therefore, H2A-S129ph may occur unimpeded when T126 is changed to an alanaine and deficient γH2A formation cannot explain the phenotype in the *hta1-T126A* mutant. In contrast, there was a reduction in H2A.1$_{T126}$ antibody recognition in the *hta1-S129A* mutant (*Figure 3E*), but given the genetic data that *hta1-S129A* does

not phenocopy the CAG instability profile of the *hta1-T126A* mutant (*Figure 3A*) we conclude that the *hta1-S129A* mutation likely disrupts the antibody epitope rather than affecting H2A.1 T126 modification.

## H2A.1 promotes fidelity of homologous recombination repair

To determine how H2A.1 and T126 contribute to fidelity of CAG repeat repair and prevent expansions, we assessed repeat stability in the absence of DNA repair pathways. If repeat expansions in the *hta1Δ* mutant are arising through a low-fidelity repair event, expansion frequency will be reduced in the absence of the relevant repair pathway.

We first tested if expansions in the absence of H2A.1 arise through NHEJ by deleting the gene encoding Lif1, a DNA ligase IV subunit, in the *hta1Δ* and plasmid *hta1-T126A* mutants. Expansion frequency remained elevated in the *hta1Δlif1Δ* and *hta1-T126A lif1Δ* double mutants (*Figure 4A and B*); therefore, instability in the absence of H2A.1 is not arising through low-fidelity NHEJ. Consistently, Moore et al found no NHEJ defects in a plasmid end-joining assay in an *hta1-T126A* mutant (*Moore et al., 2007*). We next surveyed homology-dependent repair pathways. Rad5-dependent post-replication repair can be a source of expansions during low-fidelity repair at CAG repeats (*House et al., 2014b*). Although expansions in the *hta1Δrad5Δ* double mutant are somewhat suppressed compared to the *hta1Δ* single mutant, the difference is not statistically significant (*Figure 4A*). Likewise, expansions remain elevated in the *hta1-T126A rad5Δ* double mutant (*Figure 4B*). Thus, H2A.1 does not appear to be contributing to the fidelity of post-replication repair, however a caveat to this conclusion is that *rad5Δ* elevates expansions on its own, possibly precluding observation of a suppression. Since expansions in the double mutant are less than additive and are similar to the *rad5Δ* levels, Rad5 may work upstream of H2A.1. To determine if CAG expansions in the absence of H2A.1 arise through general HR, we measured stability of the CAG repeat in the absence of two key HR proteins: Rad52 and Rad51. Expansions in the *hta1Δrad52Δ* and *hta1Δrad51Δ* double mutants are significantly reduced 2.9-fold and 5.2-fold from the *hta1Δ* single mutant, respectively (p=$2.8 \times 10^{-3}$ and p=$7.7 \times 10^{-3}$ to *hta1Δ*) (*Figure 4A*). Similarly, expansion frequencies in the *hta1Δrad57Δ* double mutant are suppressed 4.5-fold (p=$1.4 \times 10^{-3}$ to *hta1Δ*) (*Figure 4A*). Corroborating these results, expansions were also suppressed ~2 fold in the *hta1-T126A* mutant lacking either Rad52, Rad51 or Rad57, but not to the level of statistical significance due to the somewhat lower starting level of expansions in the *hta1-T126A* mutant compared to the full *HTA1* gene deletion (*Figure 4B*). Together, these results indicate that expansions in the absence of H2A.1 are arising through Rad51- and Rad52-dependent HR events and suggest that T126 plays a role in regulating HR. Since Rad57 is especially required for SCR (*Mozlin et al., 2008*), this is consistent with expansions arising during SCR, a pathway previously implicated in causing CAG instability (*Kerrest et al., 2009*; *Nguyen et al., 2017*).

## H2A.1 and H2A-Thr126 promote efficient SCR

To further evaluate the role of H2A.1 in homology-mediated repair events, we assayed the H2A mutants for their ability to undergo SCR using a genetic assay that measures rates of spontaneous unequal SCR as an estimate of overall SCR levels (*Mozlin et al., 2008*) (*Figure 4C*). In this assay, misaligned recombination between two *ade2* null alleles can result in gene conversion to a functional *ADE2* allele and the strain is converted from Trp⁺Ade⁻ to Trp⁺Ade⁺ (*Mozlin et al., 2008*). SCR is not suppressed from wild-type in the *hta2Δ* mutant, but is significantly suppressed in the *hta1Δ* mutant (2.4-fold suppression from wild-type, p=$7.9 \times 10^{-3}$; *Figure 4D*). Thus, H2A.1 is required for efficient SCR while H2A.2 is not, mirroring the differential role of the two H2A copies in CAG repeat maintenance. Similarly, SCR levels are decreased 2.7-fold from wild-type in the *hta1-T126A* mutant integrated at the endogenous *HTA1* locus (p=$4.9 \times 10^{-3}$) (*Figure 4D*). SCR is also suppressed, though more mildly, in the *hta1-S129A* mutant integrated at the endogenous *HTA1* locus (1.9-fold from wild-type; p=0.03; *Figure 4D*), in agreement with a previous report that found a mild defect in SCR during repair of a DSB in an *hta1-S129A* mutant (*Conde et al., 2009*). Rates of SCR in the *hta1-T126A* and *hta1-S129A* mutants are also reduced in the *hta2Δ* background (*Supplementary file 4*), indicating that this reduction does not depend on the presence or absence of H2A.2. Since we observe a decrease in SCR in the *hta1Δ* and *hta1-T126A* mutants but an increase in CAG expansions that is HR-dependent, we conclude that the defect in efficiency of SCR must occur after the initiation

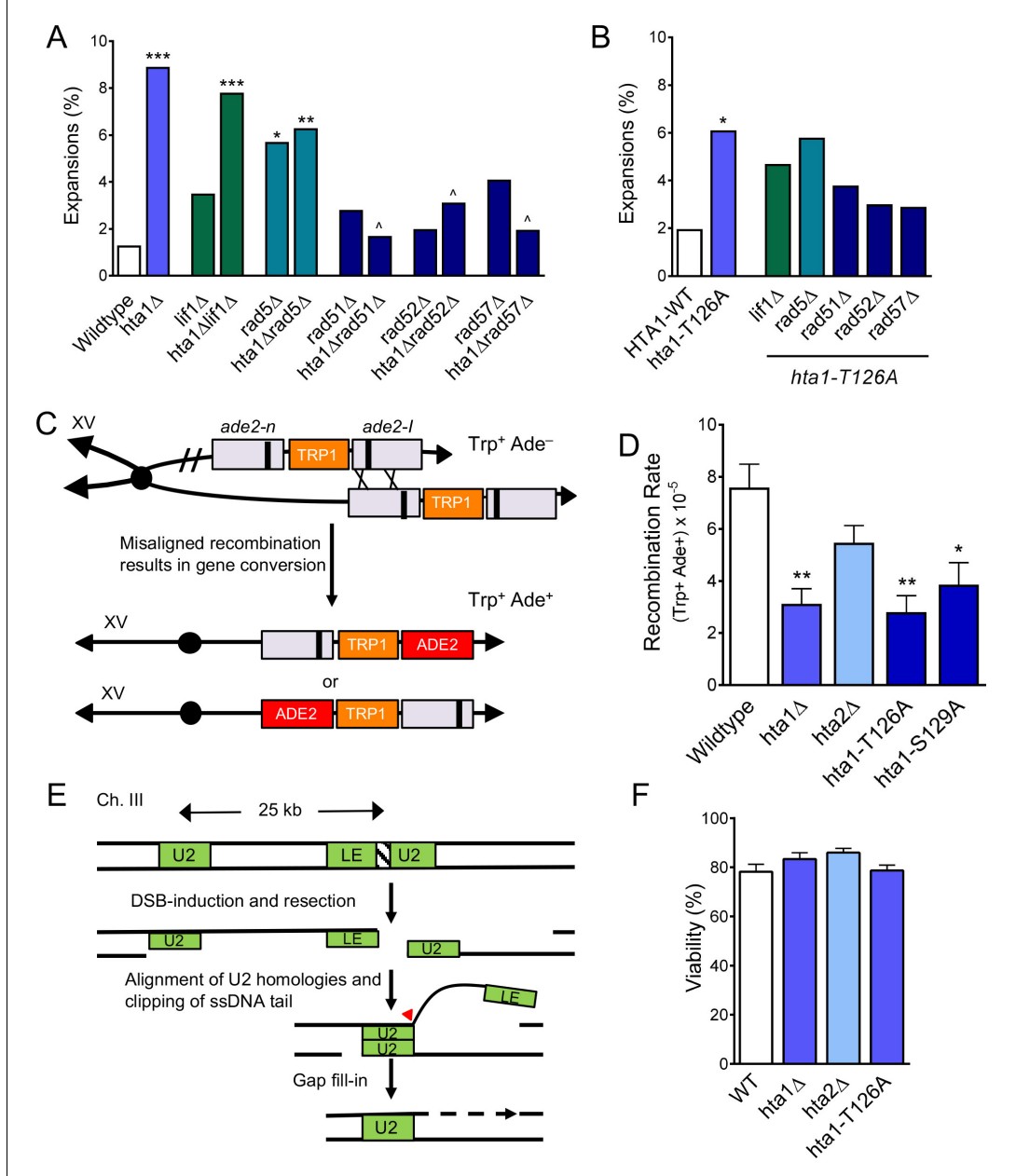

**Figure 4.** H2A.1 promotes HR fidelity to prevent (CAG)$_{85}$ expansions and is required for efficient SCR. (**A**) Repair proteins were deleted in *hta1Δ* mutant. Changes in CAG repeat length were assessed as in *Figure 1C*. Statistical deviations were calculated by Fisher's Exact Test: *p<0.05 to WT, **p<0.01 to WT, ***p<0.001 to WT; ˆp<0.05 suppression from *hta1Δ*. (**B**) Repair proteins were deleted in the strain expressing *hta1-T126A* from the plasmid; no copy of H2A.2 is present in these cells. Changes in CAG repeat length were assessed as in (**A**). Statistical deviation from the plasmid *HTA1* wild-type were calculated by Fisher's Exact Test: *p<0.05; statistical deviation from the plasmid *hta1-T126A* was also tested, but none were significantly suppressed. (**C**) Misaligned recombination during SCR can be measured by gene conversion from Trp⁺ Ade⁻ to Trp⁺ Ade⁺ (*House et al., 2014b*; *Mozlin et al., 2008*). If recombination occurs on either side of the *ade2-n* and *ade2-I* mutations (indicated by X's) between aligned mutant alleles, then gene conversion will create a WT *ADE2* gene on one of the chromatids. The *ade2-I* gene conversion resulting from the indicated cross is shown directly below the arrow; an alternate alignment and gene conversion of the ade2-n allele is also possible (bottom chromosome). (**D**) Rates of spontaneous unequal SCR. For these experiments, the *hta1-T126A* and *hta1-S129A* mutations were integrated at the genomic locus, replacing the wild-type copy of *HTA1*; *HTA2* remains intact in these strains. SCR rates for individual experiments are in *Supplementary file 5*, including for the *HTA1* point mutants with *HTA2* deleted (presence or absence of *HTA2* did not change the results). Average rate and SEM is shown. Statistical differences were calculated using a Student's t-test: *p<0.05 to WT, **p<0.01 to WT. (**E**) SSA at a DSB can be measured by viability in the presence of a galactose-induced HO DSB. Upon galactose treatment a DSB at the inserted HO cut site in the *LEU2* gene results in resection to the U2 region of homology 25 kB away. Repair occurs via single strand annealing between the U2 segments followed by cleavage of the non-homologous tail (arrowhead) and gap filling

*Figure 4 continued on next page*

*Figure 4 continued*

(dotted lines), eliminating the HO target site, and cells survive in the presence of galactose (*Vaze et al., 2002*). (F) Percent viability as measured by the number of colonies on YP-Gal divided by the number of colonies on YPD. Statistical deviation from wild-type was tested by a Student's t-test; none were significantly different from wild-type (*Supplementary file 6*).

The online version of this article includes the following figure supplement(s) for figure 4:

**Figure supplement 1.** Single strand annealing and checkpoint response.

step, such as during the D-loop synthesis. In summary, these results demonstrate that H2A.1 and H2A.1-T126 are required for efficient spontaneous SCR. This supports our conclusion that H2A.1 and H2A.1-T126 are required for proper recombination at the CAG repeat tract to prevent repeat expansions, and extends this finding to non-repetitive DNA sequences.

Histone H2A.1 could affect several steps of homology-mediate repair, including resection, homology search and invasion, D-loop synthesis, gap fill-in, or checkpoint signaling. To test which stages of repair are impacted by H2A.1, we used a single-strand annealing (SSA) system to simultaneously monitor SSA efficiency, repair kinetics, and the checkpoint response after induction of a DSB. In this system, an HO recognition site has been introduced into the *LEU2* locus and galactose-induction results in a single HO DSB at this location. At the DSB, resection occurs on both sides of the break and repair via SSA occurs between the two U2 regions of homology (*Figure 4E*) (*Vaze et al., 2002*). Repair kinetics are monitored by Southern blotting and DNA damage checkpoint activation is monitored by Western blotting for phosphorylated Rad53. Using this system, we found no difference in viability or repair kinetics between *hta1Δ*, *hta2Δ*, and wild-type strains (*Figure 4F*; *Figure 4—figure supplement 1A*; *Supplementary file 5*). This indicates that H2A.1 is not important for resection, alignment, or gap fill-in, which are required steps of repair via SSA. Similarly, we saw no significant defect in the Rad53ph checkpoint response in *hta1Δ* (*Figure 4—figure supplement 1B*), suggesting that H2A.1 is not specifically required to mediate or recover from the DNA damage checkpoint response, though some other aspect of cell cycle regulation could be affected. Since repair via SSA does not require invasion into a homologous template or D-loop extension, this result strongly suggests that H2A.1 is specifically important for homology-mediated repair requiring a D-loop.

## H2A.1 functions with Pol32 during SCR and promotes efficient break-induced replication (BIR)

Since our results indicated that H2A.1 is required during D-loop mediated repair and CAG expansions could occur during DNA synthesis, we tested the role of the Polδ subunit Pol32, which is required for Polδ processivity. Interestingly, SCR is significantly suppressed from the wild-type in the *pol32Δ* mutant. This establishes that the Pol32 subunit of Polδ is required for D-loop extension during the short tract recombination measured by this assay (less than 1 kb). Notably, SCR suppression in the *pol32Δ* mutant is epistatic to the absence of H2A.1 and a phosphorylatable H2A.1-T126 residue, as the SCR rate is not further diminished in the *hta1Δpol32Δ* or *hta1-T126A pol32Δ* double mutants (*Figure 5A*). Therefore, these results support that H2A.1 and Thr126 are important in facilitating D-loop extension by Polδ during recombination.

We next tested the role of Pol32 in CAG expansions. In the *pol32Δ* single mutant, CAG repeat expansions are significantly increased over wild-type (5.8-fold over wild-type, $p=2.0\times10^{-3}$), indicating that processive DNA synthesis by Polδ is required to prevent repeat expansions (*Figure 5B*). We also found that CAG fragility is significantly increased in the *pol32Δ* mutant (*Figure 5—figure supplement 1A*; *Supplementary file 1*). We tested whether instability in the absence of Pol32 was due to its role in normal replication or recombination-associated DNA synthesis by deleting Rad51 in a *pol32Δ* mutant (*Figure 5B*). Expansions in the *pol32Δ* mutant were suppressed in the absence of Rad51 (to 3.4%, 2.3-fold decrease from *pol32Δ*, p=0.17), suggesting that expansions are at least in part due to polymerase slippage during D-loop synthesis. The increase in expansion frequency in the *pol32Δ* mutant is similar to that in the *hta1Δ* and *hta1-T126A* mutants. However, CAG expansion frequency drops below the level of each single mutant in the *hta1Δpol32Δ* and *hta1-T126A pol32Δ*

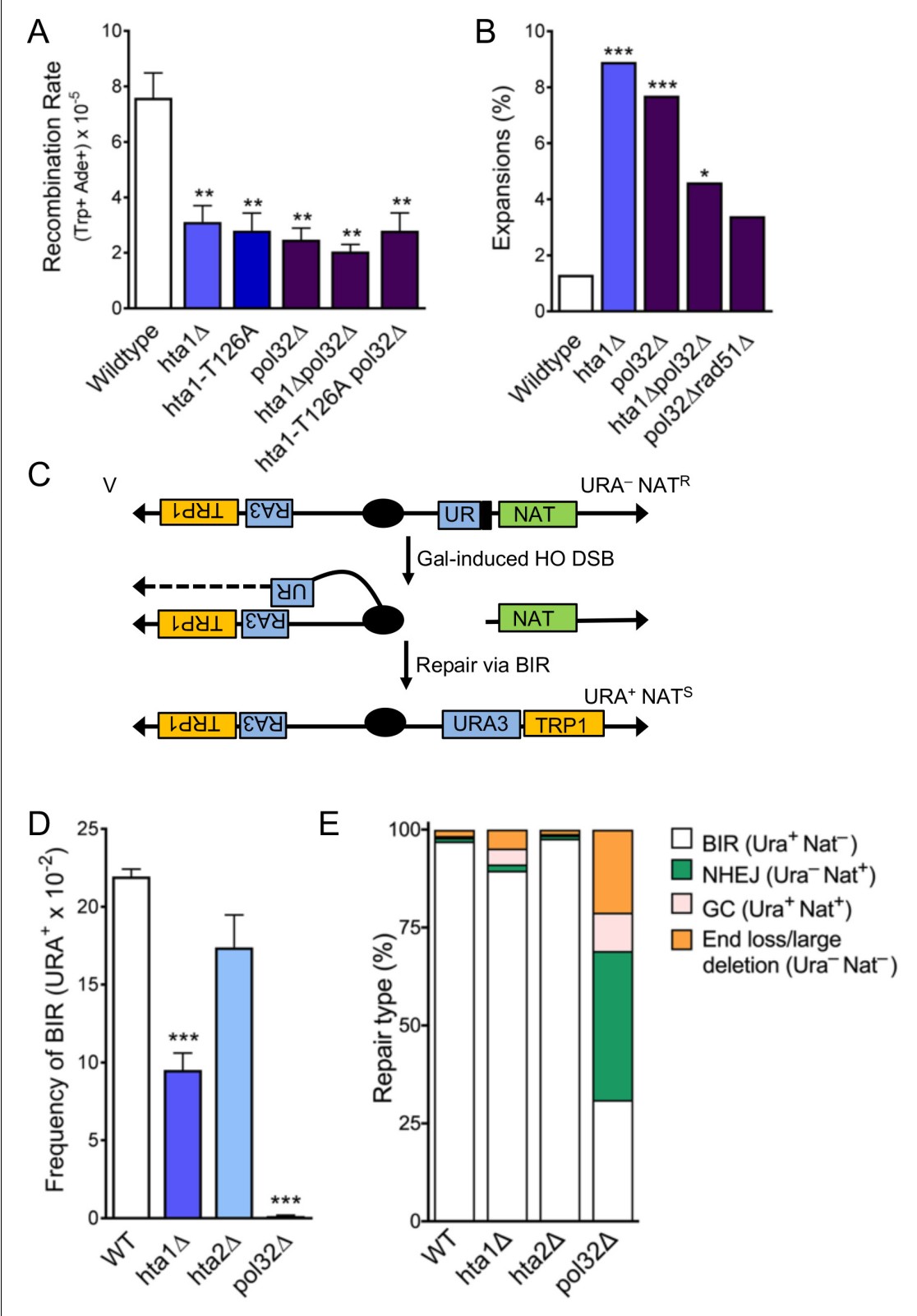

**Figure 5.** H2A.1 and Pol32 work in the same pathway during SCR, and H2A.1 is required for efficient BIR. (**A**) Rates of spontaneous unequal SCR, assessed as in *Figure 4D*. Statistical deviation from wild-type was calculated using a Student's t-test, **$p < 0.01$. (**B**) Changes in CAG repeat length were assessed as in *Figure 1C*. Statistical deviation from wild-type was calculated by Fisher's Exact Test: *$p < 0.05$, ***$p < 0.001$. (**C**) Assay to measure BIR at an HO-induced DSB. The first 400 bp of the URA3 gene (UR; right arm) is upstream of an HO cut site (stripes). Homology driven repair of the HO induced

*Figure 5 continued on next page*

*Figure 5 continued*

DSB can occur with the remaining 404 bp of the URA3 gene (A3; left arm) and if completed via BIR renders the cell URA⁺ and NAT^S (*Anand et al., 2014*). (D) Frequencies of BIR during repair of an HO-induced DSB; SEM of 4–9 replicates is shown (*Supplementary file 7*). Statistical deviation from wildtype was calculated using a Student's t-test, ***p<0.001. (E) The type of repair induced by the HO DSB was evaluated by individually scoring all colonies for growth on YC-URA and YEPD+Nat. Colonies that were URA⁺ NAT^S are were repaired via BIR. Other types of possible repair events include gene conversion (URA⁺ NAT^R), non-homologous end-joining (URA⁻ NAT^R), and de novo telomere addition (URA⁻ NAT^S). Statistical analysis of BIR frequencies and repair types can be found in *Supplementary file 7*.

The online version of this article includes the following figure supplement(s) for figure 5:

**Figure supplement 1.** CAG fragility rates in the absence of Pol32.

double mutants (*Figure 5B*; *Supplementary file 2*), suggesting that Pol32-dependent synthesis is responsible for some of the expansions occurring in the *hta1Δ and hta1-T126A* backgrounds. Though the suppression is not statistically significant (for example p=0.12 for *hta1Δpol32Δ* compared to *hta1Δ*), it is markedly reduced from the additive levels expected if there was no interaction between the pathways (for example 4.6% in *hta1Δpol32Δ* vs. 16.6% predicted for additive). Since the expansions in *hta1Δ* and *hta1-T126A* are suppressed in the absence of Rad51, Rad52, Rad57, and Pol32 (*Figures 4A* and *5B*), they are likely arising downstream of synapsis and initiation of DNA synthesis during recombination. Taken together, these results suggest that the CAG expansions observed in the absence of H2A.1 occur during a Pol32-dependent recombination process. For example, H2A.1 could promote efficient D-loop extension during replication of the donor strand, thereby preventing opportunity for DNA secondary structure formation.

Pol32 is known to be especially important for BIR, which involves extended D-loop synthesis that can proceed for many kilobases (*Lydeard et al., 2007*; *Anand et al., 2013*; *Malkova and Ira, 2013*) (see also *Figure 5D*). Considering the importance of Pol32 in preventing CAG expansions and promoting SCR and that it appears to function in the same pathway as H2A.1, we wondered if H2A.1 might also have a role in BIR. To directly test the role of H2A.1 in BIR, we used a system in which a DSB induced by the HO endonuclease can result in a non-reciprocal translocation when repair proceeds via BIR (*Anand et al., 2014*) (*Figure 5C*). BIR frequency is suppressed 2.3-fold from wild-type in the *hta1Δ* mutant (p=1.0×10⁻⁴ to WT), but remains at wild-type levels in the *hta2Δ* mutant (*Figure 5D*) and the *hta1-T126A* mutant (*Supplementary file 6*). Therefore, while H2A.1 is important for BIR, this function is either not regulated by T126 or the *T126A* phenotype was too subtle to be revealed by this assay. Interestingly, histone loss in response to DSBs has been implicated in promoting recombination and DNA repair rates (*Hauer et al., 2017*). Therefore, given that the BIR system depends on an induced DSB, altered histone levels in the *hta1Δ* mutant may increase gene conversions or other conservative repair pathways at the expense of BIR.

After HO-cutting, the cells were also surveyed for repair type by pinning colonies from the YP-Gal plates onto YPD, YPD+Nat and YC-URA. Surviving colonies were used to calculate the percent of colonies that underwent the indicated repair type (*Figure 5E*). The repair outcome profile in the *hta1Δ* mutant is altered from wild-type as a greater proportion of cells undergo other types of repair instead of BIR (*Figure 5E*) (*Supplementary file 6*). While less severe, the increase in these other types of repair events in the *hta1Δ* mutant is similar to the phenotype in the *pol32Δ* mutant, and therefore is likely a consequence of decreased BIR rates (*Figure 5E*). We conclude that deletion of H2A.1 results in decreased BIR and alterations in repair type frequencies, demonstrating that H2A.1 plays a role in facilitating efficient BIR. Together with the SCR data, this suggests that expansions in the *hta1Δ* mutant may be due to defective D-loop extension, since this step occurs during both SCR and BIR.

## Discussion

Stemming from an initial observation in a genetic screen that CAG repeat fragility and instability were elevated in an *hta1Δ* mutant but not in an *hta2Δ* mutant, we demonstrated that H2A.1 and H2A.2 differentially contribute to homology-mediated repair. The histone subtypes have been

documented before to play different functions during the *S. cerevisiae* life cycle. The absence of gene product from *HTA1* and *HTB1* led to a constitutive heat shock response after exposure to high temperature, and this phenotype was not rescued with an additional copy of *HTA2* and *HTB2* (33). Taken with our results demonstrating differential roles for the H2A copies in preventing HR-mediated CAG repeat expansions, facilitating sister chromatid recombination, and promoting BIR, this further supports that H2A.1 is specifically important for repair or recovery from DNA damage. Our data indicate that H2A.1 is more efficient at promoting high-fidelity HR than H2A.2 because of protein sequence; specifically, the position of the phosphorylatable threonine in the H2A.1 C-terminal tail is more advantageous to repair than H2A.2. Both copies of yeast H2A contain the SQEL motif in the C-terminal tail, and therefore both copies are considered homologs of mammalian H2AX. A compelling implication of our result here is that yeast H2A.1 is in fact the closer homolog of mammalian H2AX, as it plays a greater HR-specific role than H2A.2. Like H2A.1, mammalian H2AX also contains a phophorylatable threonine two residues before serine 139. Also similar to H2AX, H2A.1 is initially a smaller proportion of the total H2A pool (*Moran et al., 1990*; *Rogakou et al., 1998*). Consequently, yeast H2A.1 and H2A.2 may be more akin to histone variants than histone copies.

## A role for H2A.1-T126 in promoting high-fidelity recombination-mediated repair

H2A-Thr126 has previously been shown to be phosphorylated in vivo but its phosphorylation state after DNA damage and its overall contribution to break repair was unclear (*Wyatt et al., 2003*; *Moore et al., 2007*; *Harvey et al., 2005*; *Chambers and Downs, 2007*). Using a naturally unstable expanded CAG repeat tract, we have shown that H2A.1-T126 is important for maintaining CAG repeats and plays a role in promoting efficient SCR. Our results indicate that overall levels of HR/SCR are suppressed in the absence of H2A.1-T126, and that the recombination that does take place proceeds with low fidelity, leading to repeat expansions. Although our results implicate a requirement for phosphorylation of H2A.1-T126 in DNA repair, it is formally possible that some other physical property of threonine at position 126 may be important for repair, rather than phosphosphorylation per se. For example, the amino acid sequence in the H2A.1 C-terminal tail with threonine at position 126 may be more advantageous to DNA repair factor recruitment, independent of T126 modification status. Alternatively, H2A.1-T126 could influence neighboring modifications, such as S122 phosphorylation or acetylation of K124 or K127, though our data indicate that T126 does not modulate S129 phosphorylation.

The fact that expansions, an addition of bases, occur in *hta1Δ* and *hta1-T126A* mutants is most supportive of a role for H2A.1 in promoting a synthesis step of DNA repair. The expansions are likely arising downstream of synapsis and D-loop assembly since they were suppressed in the absence of Rad51, Rad52, and Rad57 (*Figure 4A*). Similarly, H2A.1 and T126 play no role in repair via SSA, which does not require formation of a D-loop (*Figure 4F*; *Figure 4—figure supplement 1A*; *Supplementary file 6*). Further, our data placing H2A.1 and T126 in the same pathway as Pol32 in the unequal SCR assay supports a role during D-loop extension. Therefore, we conclude that H2A.1 is most likely promoting efficient D-loop extension during replication of the donor strand. A second possibility is that H2A.1-T126 is required to promote a later step in the process such as D-loop resolution or re-establishment of the chromatin structure of the repaired gap. This could explain why the H2A.1-T126A mutant did not have a discernable effect on BIR, which does not include re-engagement of the extended D-loop with the initiating DNA molecule.

Our data indicated that the role of H2A.1-T126 was not directly dependent on S129ph, however, the two residues could act together to affect an outcome. H2A-S129ph begins to accumulate at the CAG repeat 20 min into S-phase and peaks at 40 min (*House et al., 2014b*). H2A.1 is either incorporated or phosphorylated at the CAG tract 40 min into S-phase, appearing only after initial accumulation of S129ph but coinciding with peak S129ph. Specific incorporation of H2A.1 at the CAG repeat when DNA damage is occurring is highly suggestive of some role for T126 at that time. Since our antibody also detects phosphorylated H2A.1, it could be that the additional ChIP signal is due to existing H2A.1 becoming phosphorylated locally. Alternatively it may reflect damage-specific incorporation of H2A.1, which may or may not be phosphorylated on T126 (*Figure 6A*). Levels of H2A.1-T126 decreased upon MMS damage (*Figure 3E*), and whole-genome proteomics did not identify H2A-T126ph as a DNA damage-inducible modification (*Bastos de Oliveira et al., 2015*; *Smolka et al., 2007*; *Zhou et al., 2016*), though because T126 is surrounded by two lysines (K124,

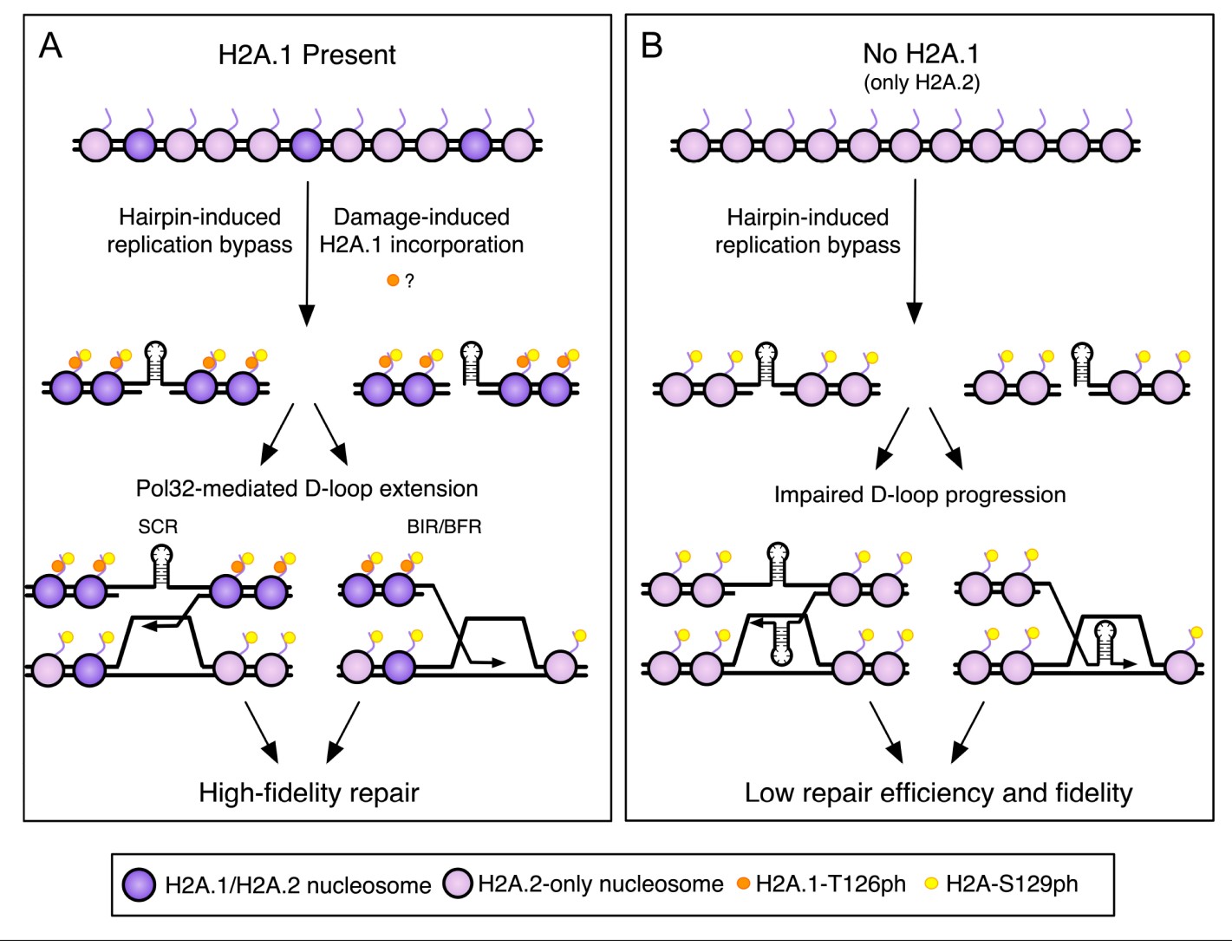

**Figure 6.** A model for high-fidelity HR-mediated repair promoted by H2A.1 and T126 phosphorylation. The initiating lesion at the CAG repeat could be a nick, gap, or a one-ended break (e.g. created at a replication fork). (A) Incorporation of H2A.1 at sites of damage (with or without T126ph; with T126ph is shown) promotes repair fidelity. The presence of H2A.1 will promote efficient D-loop extension after invasion into the sister chromatid, leading to repair with fidelity and maintenance of CAG repeat number. Alternatively, the resolution of recombination intermediates may require H2A.1 to prevent multiple reinvasion events that could lead to repeat expansions. Though T126ph is shown as present during BIR/BFR for continuity with the previous diagram, our data suggest that the T126 identity has minimal effect on this step in the context of BIR (BFR was not tested). See discussion for details. (B) Without H2A.1 only H2A.2 with T125 is present and D-loop extension is impeded, allowing hairpins or misalignment during Rad51-dependent invasion, leading to less efficient repair and CAG repeat instability. Purple nucleosomes contain H2A.1, pink nucleosomes are H2A.2-containing only. Orange dots represent H2A.1-T126ph, which could either be a pre-existing modification or induced locally at the damage site. Yellow dots represent H2A-S129ph.

K127), it will end up on a peptide only three amino acids long after trypsin digest, which will not be easily detected by mass spectrometry. Nonetheless, we favor a model in which T126ph is a pre-existing modification independent of S129ph, but that these two modifications may work together to promote efficient repair, for example by binding or excluding certain repair factors in a combinatorial fashion. In this scenario, T126ph would become enriched at the site of damage because of damage-specific incorporation of H2A.1 during repair (*Figure 6*) and disappear after repair is complete, either by removal of H2A.1 or local dephosphorylation of the T126 residue. Depending on the

phosphorylation status of both residues, a repair protein could recognize one or both residues, which could facilitate removal of S129ph-bound proteins or turnover of S129ph/T126ph-containing nucleosomes, influencing the overall progression of the repair process. This idea is supported by the similar increase in expansions in the *T126A* and *T126E* mutants, suggesting that dephosphorylation of T126 could be important for repair fidelity. Alternatively, the presence of a threonine three amino acids away from S129ph, independently of its modification state, could influence repair, or the T126 effect may be independent of S129ph, even though they occur coincidentally.

Since we found that H2A.1 is specifically important for D-loop mediated HR repair, ATPase chromatin remodelers are attractive candidates for direct interaction with H2A.1-T126 or H2A.1-T126ph. Interestingly, in G2/M when a sister chromatid is available for recombination, γH2A is dispensable for recruitment of several chromatin modifiers, including Ino80 and subunits of RSC, NuA4, Rpd3, SWI/SNF, and SWR-C (*Bennett et al., 2013*). H2A.1-T126-mediated recruitment of chromatin remodelers could be important for opening the chromatin structure to allow access by repair factors, remodeling chromatin on the donor strand to facilitate synapsis or D-loop extension, or resetting the chromatin to promote repair resolution. The *hta1-T126A* mutant displays a defect in telomere positioning effect (TPE) (*Wyatt et al., 2003*), consistent with a role in reestablishing chromatin structure after repair or replication. We previously found that RSC and NuA4 are required to promote high-fidelity repair of the CAG repeat, and showed that RSC subunits are recruited to the repeat during S-phase (RSC2 peaks at 20 min, RSC1 at 40 min). We concluded that these factors are promoting post-replication repair events (*House et al., 2014b*), however, it is possible that RSC and NuA4 also have a more general role in any D-loop mediated repair process. Recruitment of any of these proteins to the CAG repeat during repair could facilitate chromatin remodeling to promote efficient D-loop progression or resolution and high-fidelity repair.

## A model for H2A.1 in HR-mediated repair of CAG repeats

Hairpins formed by CAG or CTG repeats interfere with replication and induce joint molecules between sister chromatids, which have been visualized by two-dimensional gel electrophoresis (*Kerrest et al., 2009*; *Nguyen et al., 2017*). We propose a model in which gaps caused by replication bypass of the CAG repeat are repaired via SCR, and that incorporation of H2A.1 and/or H2A.1-T126ph ensures efficient remodeling and progression of Pol32-mediated D-loop extension, leading to high-fidelity repair (*Figure 6A*). In the absence of H2A.1, suboptimal signaling from the H2A.2-T125 residue may inefficiently recruit or retain chromatin remodelers or other DNA repair proteins, impeding D-loop extension due to a chromatin state that is not permissive to extension and copying (*Figure 6B*). Inefficient progression of the recombination intermediate could lead to Polδ stalling or transient dissociation of the 3' end of the invading strand, which would allow opportunity for CAG repeat secondary structure formation and slippage during synthesis, leading to repeat length changes in the repaired DNA strand. Alternatively, chromatin remodelers or repair factors influenced by the presence of H2A.1-T126 may be required to reset the chromatin structure after repair. The permissive chromatin state could allow multiple, aberrant invasion events that would increase the opportunity for misalignments during D-loop initiation or elongation, resulting in repeat instability.

Perhaps less commonly, a replication fork stalled by a CAG or CTG hairpin may be converted to a one-ended DSB that could facilitate HR-dependent fork restart, similar to BIR (e.g. broken fork repair or BFR; *Malkova and Ira, 2013*). If fork restart proceeds with low fidelity, such as if recombination structures are misaligned or hairpins cause strand slippage, mutations can arise. Our data suggest that H2A.1 is required for efficient BIR, and CAG expansions occurring in the *hta1Δ* background were reduced when Pol32 was deleted, inhibiting BIR. Indeed, recovery from broken forks via BIR/BFR has recently been proposed to cause large-scale expansions of a $(CAG)_{140}$ repeat tract in yeast (*Kim et al., 2017*), and our data at the $(CAG)_{85}$ repeat are consistent with these results. However, in the wild-type background, expansion frequency is significantly increased in a *pol32Δ* mutant where BIR is suppressed (*Figure 5B*), indicating that BIR is not the only pathway creating expansions. Our data show that at least some expansions in the *pol32Δ* mutant occur during recombination (*Figure 5B*), suggesting that efficient D-loop synthesis through a CAG repeat requires Pol32. It remains possible that some expansions occurring in the *pol32Δ* mutant are arising by a different mechanism, such as impaired Polδ synthesis during replication. While efficient BIR was dependent on H2A.1, the rates were not affected in the *hta1-T126A* mutant, thus, the requirement of H2A.1 during BIR is not through the modifiable T126. The BIR assay involves a DSB whereas the

spontaneous SCR assay and CAG repeat expansion assay is measuring the response at endogenous DNA lesions that will include a mix of nicks, gaps, and a small proportion of DSBs. We note that CAG fragility, CAG contractions, and BIR, that all involve an initiating DSB, were dependent on H2A.1 but not the T126 residue. Therefore, our data are consistent with H2A.1 levels affecting DSB repair while the H2A.1 sequence-specific role is during gap repair/SCR.

## Conclusions

At the occurrence of DNA damage, recombination is thought to be more protective to genome integrity than end joining because the repair is templated. However, recombination can itself be mutagenic if it does not proceed in a regulated manner (*Polleys et al., 2017*; *Guirouilh-Barbat et al., 2014*). Turnover of chromatin modifications during repair is an attractive model for facilitating proper repair progression, either by influencing chromatin reorganization (reviewed in *Polo, 2015*) or by facilitating sequential recruitment and release of repair factors (reviewed in *House et al., 2014a*; *Price and D'Andrea, 2013*). Our results demonstrate a genetic interaction between H2A.1 and Pol32 (Polδ) in maintaining CAG repeat stability. The timing and reading of H2A.1-T126 and other chromatin marks may determine how effectively the Polδ complex moves through the donor strand during repair, ensuring that repair is efficient (timely) and that it proceeds with high-fidelity, limiting mutagenic repair outcomes.

Our results suggest that yeast H2A.1 is a closer homolog of mammalian H2AX, whereas H2A.2 is more functionally equivalent to mammalian H2A. Although the density of nucleosomes can vary at different genomic loci, the abundance of histones throughout the genome means they are readily available at the frontlines of DNA damage. In yeast, histone H2A.1 and H2A.2 are both present in a healthy cell and are likely interspersed throughout the genome. Transcript from H2A.2 is present in higher proportion than H2A.1 under normal conditions (cited in *Moran et al., 1990*), but there must be an adequate amount of H2A.1 present on the chromatin to act in a fashion that promotes DNA repair. This may be similar to H2AX distribution in human cells, which is 2.5–25% of the total H2A pool (*Rogakou et al., 1998*). Further, in mammalian cells the H2AX histone variant is also required for efficient SCR (*Xie et al., 2004*). The analogous H2AX threonine (T136) is also phosphorylated in mammalian cells (*Li et al., 2010*; *Bennetzen et al., 2010*). A T136V mutation did not decrease survival after IR or DSB-induced HR in mouse cells, but gap repair and repair fidelity were not analyzed (*Xie et al., 2010*). Thus, it would be interesting to test whether mammalian H2AX T136 plays a role in repair fidelity analogous to the role described here for yeast H2A.1-T126.

## Materials and methods

In all cases, reference to independent experiments refers to independent biological replicates, which were done from separate starting colonies or cultures that were treated separately (e.g. plated for analysis or prepared separately for DNA or protein preparation).

### Yeast strain construction

The yeast strains used in this study are described in *Supplementary file 11*. *Mutant strain construction:* Genes were deleted via PCR-based gene replacement with selectable gene markers. Integration of the selectable marker was verified using PCR at the 3' and 5' integration junctions and primers internal to the target gene to verify ORF absence. All assays were done with at least two independently created strains.

### CAG repeat fragility assays

Assays to measure CAG repeat fragility were performed by fluctuation test as previously described (*House et al., 2014b*; *Polleys and Freudenreich, 2018*). Briefly, 10 single colonies with verified (CAG)$_{85}$ tract lengths were grown 6–8 divisions and *URA3* marker loss was tested by plating on FOA-Leu. Mutation frequency was calculated by the Method of Maximum Likelihood (*Hall et al., 2009*). Rates were evaluated for significant deviation from WT by the student's t-test, as we were interested in comparison to the wild-type rate. At least three biological replicates were performed for each strain, using at least two independent transformants. See *Supplementary file 1* for a list of individual assays. Outliers according to the Grubb's test were removed.

## CAG repeat stability assays

PCR-based stability assays were performed as described (*House et al., 2014b*; *Polleys and Freudenreich, 2018*). Briefly, single colonies with verified (CAG)$_{85}$ tract lengths were grown 6–8 generations and the tract length was measured in ~100–300 daughter colonies from two independent transformants. PCR products were run in 2% MetaPhor agarose or a custom gel mix on a fragment analyzer (Custom kit DNF945, Advanced Analytical Technologies, Inc). The number of expansions and contractions were evaluated for significant deviation from wild-type using the Fisher's Exact Test (see *Supplementary file 2*). This test is appropriate as we are comparing categorical data (for example # expansions in wt vs # expansions in mutant) in small sample sizes (e.g. less than 1000). Sample size was determined by the number needed to determine statistical significance balanced by the practicality of the number of PCR reactions that could be done with available resources.

## Chromatin analysis by MNase digestion and indirect End-labeling

Chromatin digestion was performed as previously described (*Koch et al., 2018*, adapted from *Wu and Winston, 1997*), with the following modifications: spheroblasts were digested with 0, 0.25, 2.5, or 7.5 units of MNase and the DNA pellet RNase A digested for 30 min. The DNA was extracted twice using an equal volume of chloroform and precipitated with $NH_4OAc$ and isopropanol. *Southern Detection:* MNase digested DNA (20–30 µg) was run in 1.5% agarose at 80V for 6 hr and Southern blotted as previously described (*Koch et al., 2018*). Chromatin structure was probed with a $^{32}$P labeled 358 bp PCR fragment amplified from 102 bp upstream of the CAG repeat on YAC CF1. The experiment was repeated multiple times and a representative blot is shown.

## Mononucleosome positioning detection by Illumina bead array

Chromatin was isolated and MNase digested as described above, except mononucleosomes were prepared by digesting the chromatin with 10 units of MNase for 15 min. Purified mononucleosomal DNA was amplified using the GenomePlex Whole Genome Amplification kit (Sigma). Amplification and ~150 bp fragment sizes were verified by running in 1.5% agarose. Amplified mononuclesomes were purified with a GenElute PCR clean-up kit (Sigma) and 3' biotin end-labeled (Pierce); DNA was chloroform extracted, and labeling was verified by dot blot according to the manufacturer's instructions. Purified mononucleosomes were applied to a custom Illumina array that contained YAC CF1 sequence spanning a region 425 bp upstream to 438 bp downstream of the repeat tract in 30 bp non-overlapping probes (*Supplementary file 3*). For each sample, a 12 µl aliquot of the purified mononucleosomal DNA was mixed with 3 µl of 15.3X SSC buffer containing 2.4% SDS, heated at 95° C for 5 min, and snap cooled. The sample was applied to the sample zone of the microarray and hybridization was carried out overnight at 62°C with humidity. Arrays were washed in stringency wash (0.2% SDS and 2X SSC buffer) with rocking for 5 min, followed by 0.05X SSC buffer with vigorous agitation for 1 min. To reduce non-specific binding of the stain, arrays were incubated with blocker casein in 1X PBS (Thermo) for 5 min. Arrays were then stained for 10 min in blocker casein with 1 µg/ml streptavidin-Cy3 in PBS (Invitrogen). Chips were vigorously agitated in 0.05X SSC buffer for 1 min, dried with compressed air, and scanned with a BeadArray Reader (Illumina) using Direct Hybridization settings with a factor of 5. Signal intensity was exported to Excel via BeadScan. The experiment was repeated twice for WT and 3x for *hta1Δ* and *hta2Δ* strains.

## H2A sequence alignments

All DNA sequences were acquired from the Saccharomyces Genome Database (https://www.yeastgenome.org/) and aligned using SerialCloner or SnapGene. All protein sequences were acquired from Uniprot (http://www.uniprot.org/) and aligned by Clustal Omega (http://www.ebi.ac.uk/Tools/msa/clustalo/).

## Western blotting of H2A

Strains were grown in YPD at 30°C with agitation to log phase (OD$_{600}$ = 1); cells were either untreated or exposed to the following drug treatments at 30°C with agitation: 0.01% MMS for 2 hr, 0.03% MMS for 1 hr, 0.2M HU for 1 hr, 0.2M HU + 0.03% MMS for 1 hr. Lysates were extracted according to *Adams et al. (1997)* and https://research.fhcrc.org/gottschling/en/protocols/yeast-protocols/protein-prep.html and Western blotted onto PVDF. Blots were probed with anti-H2A (Abcam

#13923; 1:5000), anti-H2A$_{T126}$ (Aves Lab custom antibody; gift from Krebs lab; 1:2500), or anti-H2A$_{S129ph}$ (Abcam; ab15083; 1:2500) in 2.5% milk in 1X PBS (pH 7.4). The signals were detected with HRP-conjugated secondary antibody (1:2500) and ECL (Pierce). Western blot signals were quantified by ImageJ. The fold change in signal from WT was determined by comparing the relative quantification value (rqv) of a mutant to the relative quantification value of the WT (rqv = ratio of the indicated band to the loading control, with background subtracted). Each experiment was repeated at least three times and quantified; individual and mean values, SEM and p-values to control are reported in *Supplementary file 9*.

## H2A$_{T126}$ antibody information

The anti-H2A.1$_{T126}$ custom chicken antibody was generated by Aves Lab (http://aveslab.com). The antibody was raised against the phosphopeptide KKSAKA[pT]KASQEL. An attempt to create a second batch of this antibody by the same procedure was not specific, thus the specificity of antibodies created in this manner is variable. Only experiments with the H2A.1-specific antibody preparation are included.

## Chromatin immunoprecipitation

ChIP was performed in biological duplicate as previously described (*Aparicio et al., 2005*; *House et al., 2014b*) using anti-H2A.1$_{T126}$ (custom preparation, Aves Lab), normal chicken IgY (Millipore AC146), anti-H2B (Abcam ab1790), or anti-H3 (Millipore 05–928) and Dynabeads Protein G (Invitrogen) or anti-IgY agarose beads (Invitrogen) for immunoprecipitation. Time points were taken after release from synchronization in 5 µM α-factor for 1.5 hr at 30°C with shaking. DNA levels were measured by qPCR using SYBR Green PCR mastermix (Roche) or Power SYBR Green PCR Master Mix (Applied Biosystems); qPCR reactions were run in duplicate amplifying a 200 bp fragment 0.6 kb upstream of the CAG repeat or at *ACT1*. Enrichment at the CAG repeat was determined by the ΔΔCt method; H2B and H3 ChIP was additionally normalized to enrichment at an *ACT1* control locus. See *Supplementary file 4* for raw IP/Input values, qPCR conditions, and primer sequences.

## Sister chromatid recombination assays

Assays were performed as previously described (*House et al., 2014b*; *Mozlin et al., 2008*). Briefly, Trp+ Ade- cells were grown in YEPD to saturation. Total viable cell count was measured by plating $10^{-5}$ dilutions on yeast complete (YC) media and recombinants were selected by plating $10^{-2}$ dilutions on YC-Trp-Ade. Recombination rates were calculated by the method of the median and rates were tested for statistical significance using the Student's t-test (*Supplementary file 5*). Outliers according to the Grubb's test were removed.

## Break-induced recombination assays

Assays were modified from *Anand et al. (2014)*. Briefly, colonies on YEPD+Nourseothricin were serially diluted and plated on YEPD and YEP+Galactose in duplicate. Colonies were counted and percent viability on YEP+Galactose was determined by dividing by the number of colonies on YEPD. To determine the frequency of BIR and other types of repair, all YEP+Galactose colonies were pinned onto YEPD, YEPD+Nourseothricin, and YC-URA. BIR frequency was calculated as number of URA+ NAT- cells divided by the number of colonies on YEPD and tested for statistical significance using the Student's t-test (*Supplementary file 7*).

## Acknowledgements

The authors thank L Symington for the SCR assay system, R Anand and J Haber for the BIR assay system, J Haber for the SSA assay system, J Downs for the plasmid H2A point mutant strains, Ryan Hayman and David R Walt for help designing and running the customized Illumina BeadArray, and Danae Schulz for performing the intial screen that identified H2A.1 as a candidate for further study. This work was funded by a National Institutes of Health award to CHF (Project 4 of P01GM105473, James Haber, PI), and an American Cancer Society–Ellison Foundation Postdoctoral Fellowship PF-18-125-10-DMC to EJP.

## Additional information

### Competing interests

Cailin E Joyce: is affiliated with Agenus Inc. The author has no financial interests to declare. Oliver Takacsi-Nagy: is affiliated with ArsenalBio. The author has no financial interests to declare. The other authors declare that no competing interests exist.

### Funding

| Funder | Grant reference number | Author |
|---|---|---|
| National Institutes of Health | P01GM105473 | Catherine H Freudenreich |
| American Cancer Society | PF-18-125-10-DMC | Erica J Polleys |

The funders had no role in study design, data collection and interpretation, or the decision to submit the work for publication.

### Author contributions

Nealia CM House, Conceptualization, Formal analysis, Supervision, Validation, Investigation, Visualization, Methodology; Erica J Polleys, Conceptualization, Formal analysis, Supervision, Funding acquisition, Validation, Investigation, Visualization, Methodology; Ishtiaque Quasem, Validation, Investigation, Visualization; Marjorie De la Rosa Mejia, Cailin E Joyce, Oliver Takacsi-Nagy, Data curation, Investigation, Writing - review and editing; Jocelyn E Krebs, Resources, Data curation, Writing - review and editing; Stephen M Fuchs, Resources, Supervision, Methodology; Catherine H Freudenreich, Conceptualization, Resources, Formal analysis, Supervision, Funding acquisition, Validation, Visualization, Methodology, Project administration

### Author ORCIDs

Catherine H Freudenreich (iD) https://orcid.org/0000-0002-1652-2917

### Decision letter and Author response

Decision letter https://doi.org/10.7554/eLife.53362.sa1
Author response https://doi.org/10.7554/eLife.53362.sa2

## Additional files

### Supplementary files

• Supplementary file 1. (CAG)0 and (CAG)85 fragility assay data.
• Supplementary file 2. (CAG)85 repeat instability assay data.
• Supplementary file 3. Illumina Array probe sequences.
• Supplementary file 4. ChIP data.
• Supplementary file 5. Sister chromatid recombination assay data.
• Supplementary file 6. Single strand annealing assay data.
• Supplementary file 7. Break-induced replication assay data.
• Supplementary file 8. HTA2 gene levels by qPCR.
• Supplementary file 9. Western blot quantifications.
• Supplementary file 10. Pol32 mRNA levels.
• Supplementary file 11. Yeast strains used in this study.
• Transparent reporting form

### Data availability

All data generated or analyzed during this study are included in the manuscript and supporting files.

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

**Appendix 1**

## Supplemental methods

### *HTA1* point mutants

FY406 H2A point mutant strains were a gift from Dr. Jessica Downs (**Harvey et al., 2005**). The H2A point mutants are contained on a CEN plasmid containing either *hta1-T126A* or *hta1-S129A* point mutation, a wild-type copy of *HTB1*, and a HIS3 marker. Both the *HTA1/HTB1* and *HTA2/HTB2* loci are deleted in these strains so the sole copies of H2A and H2B are expressed from the plasmid. A YAC containing a $(CAG)_{85}$ repeat tract was introduced by cytoduction as previously described (**Callahan et al., 2003**). For genomic integration of *hta1* point mutants, the *HTA1* gene plus 200 bp upstream and 347 bp downstream of the gene was cloned into either the pFA6a-KanMX6 or the pFA6a-HPH vector. The *hta1-T126A, T126E, S129A or T126A/S129A* point mutations were integrated into the *HTA1* plasmid via cloning and PCR-based gene replacement methods integrated the mutations, confirmed by PCR and sequencing. *Replacement of HTA2 in HTA1 gene locus*: A selectable KANMX6 gene was knocked-in 150 bp downstream of the HTA2 stop codon. The entire *HTA2* gene and KANMX6 marker was PCR amplified with primers with 40 bp tails homologous to the *HTA1* locus. Integration of *HTA2*+KANMX6 into the HTA1 gene locus was confirmed by PCR and sequencing. The *HTA2* gene at its endogenous locus was deleted.

### HTA1 and HTA2 gene copy number verification

*hta1Δ* and *hta2Δ* strains used in this study (**Supplementary file 8**) were grown overnight and subjected to DNA extraction via phenol:chloroform and bead beating. Samples were diluted and total genomic *HTA1 and HTA2* levels were measured in duplicate by qPCR using Power SYBR Green PCR Master Mix (Applied Biosystems, 4367659). Absolute quantities were determined by comparison to a standard curve and normalized to the control locus *ACT1* (**Supplementary file 8**). Given the high sequence similarity between *HTA1* and *HTA2*, primers were designed such that they were specific to *HTA1* or *HTA2*. Primer sequences are in **Supplementary file 8**.

### Histone Phosphatase Assay

Cells were grown to mid-log phase and subsequently treated with 0.035% MMS for 3 hr. For on blot assays, $9.25 \times 10^7$ were collected. For in tube assays $1.26 \times 10^9$ cells were collected. *CIP treatment on-blot:* Whole cell lysates were extracted according to **Adams et al. (1997)** and https://research.fhcrc.org/gottschling/en/protocols/yeast-protocols/protein-prep.html and Western blotted onto PVDF. Post-transfer, membranes were activated, blocked in 5% BSA in 1X TBST for 1 hr and washed with 1X TBS. Membranes were cut in half, placed in 5 mL of 1X Cutsmart buffer and either treated with to 10 U of calf intestinal alkaline phosphatase (CIP) (NEB #M0290S) or received no treatment. Membranes were incubated for 1 hr at 37°C. *CIP pre-treatment:* Histones were acid extracted according to **Jourquin and Géli (2017)**. For the phosphatase treatment, 0.5 ul of purified histones was resuspended in 1X Cutsmart buffer and incubated with 3 U CIP (NEB #M0290S) for 1 hr at 37°C. Control samples were incubated with no CIP added. Samples were Western blotted onto PVDF and subsequently probed. *Visualization and quantification:* Blots were probed with anti-H2A (Abcam ab13923; 1:5000), anti-H2A$_{T126}$ (Aves Lab custom antibody; gift from Krebs lab; 1:2500), or anti-H2A$_{S129ph}$ (Abcam; ab15083; 1:2500) in 5% BSA in 1X PBS (pH 7.4). The signals were detected with HRP-conjugated secondary antibody (1:2500) and ECL (Pierce). Western blot signals were quantified by ImageJ.

## Single strand annealing assays

Assays were modified from *Vaze et al. (2002)*. For viability assays, colonies were grown for 2–3 divisions in YP-Lactate at 30℃, serially diluted, and plated on YEPD and YEP+Galactose in duplicate. Colonies were counted and percent viability was determined (number of colonies on YEP+Galactose divided by the number of colonies on YEPD). See *Supplementary file 6*.

## HO induction and measuring repair kinetics

Time course experiments were performed as previously described (*Vaze et al., 2002*). Briefly, 400 mL YP-Lactate was inoculated with $1-3 \times 10^6$ cells. Cells were grown overnight at 30℃. Fifty millilitres of cells were removed and then galactose was added to a final concentration of 2% to induce the DSB. Aliquots were removed at the indicated timepoints and DNA was extracted using phenol:chloroform and bead beating. Purified DNA was normalized using a Qubit (dsDNA BR kit; Q32850), digested overnight with KpnI and Southern blotted using a probe specific to *LEU2*.

## Western blotting for phosphorylated Rad53

Protein extraction was performed as previously described (*Foiani et al., 1994*). Briefly 5 OD of cells at the indicated timepoints were treated with 20% trichloroacetic acid and bead-beating. Total protein extracts were separated on a 7.5% gel, blotted onto PVDF and probed with anti-Rad53 (Abcam ab166859; 1:1000). Phosphorylated Rad53 is visible as a retarded band on the blot.

## qPRT-PCR of Pol32 levels

mRNA extraction was done on approximately $1.2 \times 10^8$ logarithmically growing cells using the Illustra RNAspin Mini kit (GE, 25-0500-70). cDNA synthesis was done using SuperScript II Reverse Transcriptase (Invitrogen, 18064022). mRNA was diluted and POL32 levels were measured in duplicate by qPCR using Power SYBR Green PCR Master Mix (Applied Biosystems, 4367659). Absolute quantities were determined by comparison to a standard curve of genomic DNA and normalized to the control locus *ACT1* (*Supplementary file 10*). Primer sequences are in *Supplementary file 10*.

