## [Decision Letter]

**Acceptance summary:**

The role of chromatin in homologous recombination is poorly understood, and this work makes the surprising observation that in budding yeast histone H2A.1 but not H2A.2 is required for Rad51-dependent recombination. The authors are able to ascertain that the T126 in HTA.1, which is unique to H2A.1, is critical for this function.

**Decision letter after peer review:**

The previous reviews from another journal have been considered and evaluated along with the manuscript by an expert Reviewing Editor, who has the following comments:

The previous reviews and responses by the authors address the concerns raised in two rounds of review at another journal. I disagree with the previous editorial decision that the issues surrounding the H2A.1T126 anti-serum jeopardize the main conclusion of the study. The authors appropriately describe the limitations of this anti-serum and carefully interpret the data accordingly. I concur with the reviewers, who after 2 rounds of revisions, support publication of this manuscript.

The only change requested is an edit in the Significance Statement. Please change 'error-free' to 'high-fidelity' or similar. Work in the Strathern and Haber laboratories demonstrated that DSB repair by homologous recombination is associated with a significant increase in mutagenesis.

---

## [Author Response]

[Editors' note: we include below the reviews that the authors received from another journal, along with the authors’ responses.]

My co-authors and I would like to submit the manuscript “*Saccharomyces cerevisiae* H2A copies differentially contribute to recombination and CAG/CTG repeat maintenance, with a role for H2A.1 threonine 126” for consideration for publication as a Research Article in *eLife*. This manuscript was previously favorably reviewed at another journal, but ultimately rejected by the editors due to a difference in opinion about using an antibody with less than 100% specificity (but the only one available) in one of the figures. Our results are not dependent on this antibody, though they add to the genetic data, and the reviewers thought our interpretations were appropriate and the experiment using the antibody was a significant addition to the paper. In yeast, there are 2 copies of histone H2A (H2A.1 and H2A.2) that were previously thought to have identical functions. In this manuscript, we characterize a repair-specific role for H2A.1 that is not shared by H2A.2, and can be attributed to a phosphorylatable threonine residue (T126) in the C-terminal tail. We find that H2A.1 is required for efficiency of sister chromatid recombination and for promoting repair fidelity during homologous recombination. Thus, yeast H2A.1 is a closer homolog of mammalian H2AX, which is specifically required for DNA damage repair. This is the first report of this H2A.1-specific function and thus will be of great interest to the readers of *eLife*. This repair-specific function of H2A.1 will now need to be considered when thinking about the role of histone H2A in the cell.

In the text, we have remedied the typos and nomenclature/genotype concerns, added an explanation of the statistics and more consistently reported the comparisons and p-values within the text instead of relying on the supplemental table. We have also added more supplemental tables that contain all of our raw data. Point-by-point responses can be found in-line below. We thank the editor and the reviewers for your comments and suggestions, as addressing them has further strengthened our story and solidified our conclusions.

Experimentally we have added the following new findings and controls:

– Quantification of all blots, with repetitions and error bars. This allowed us to determine the reactivity of the H2A.1T126 antibody with H2A.1 versus H2A.2 variants. The specificity for H2A.1 (copy 1) is about 70%, which is in line with other histone tail antibodies on the market.

– A phosphatase treatment to show at what level the H2A-T126ph antibody preferentially recognizes the phosphorylated form of H2A.1 in vivo. We found the antibody does not discriminate between the two forms very well in vivo, showing only a 30% preference for the phosphorylated form. Therefore, we have modified our conclusions from the Western blots regarding phosphorylation accordingly.

– ChIP of the histones at the CAG repeat, showing specific enrichment of the H2A.1 variant at the same time γH2AX is maximal, while overall histone/nucleosome levels remain unchanged.

– qPCR of HTA gene levels in the *hta1Δ* and *hta2Δ* mutants verify no gene amplification at the genomic level in the strains used in this study.

– Further verification that H2A.1 and T126ph is not induced by MMS, HU, or HU+MMS (quantification of the Western blots using the H2A.1T126 antibody) and spot assays that show no sensitivity to DNA damaging agents in the hta1Δ or hta1-T126A mutants.

– RT-PCR data showing that levels of Pol32 transcript are not changed in the *hta1Δ* mutant, verifying that the *pol32Δ* phenotypes are not an indirect consequence of Hta1 affecting POL32 transcription levels.

– CAG fragility and BIR assays using the hta1-T126A mutant (genomic integration; showed no phenotype).

– An SSA assay that demonstrates no role for HTA.1 in a repair reaction that does not involve a D-loop.

– A Rad53ph Western that shows no significant checkpoint recovery defect in the *hta1Δ* mutant. The SSA and Rad53ph data allowed us to narrow down the step at which H2A.1 is most likely acting (D-loop synthesis).

Our overall conclusions have remained the same since our initial submission of the paper. We conclude that H2A.1 specifically contributes to maintaining fidelity of homology-mediated repair, and our ChIP results suggest repair-specific incorporation of this histone variant. Further, we demonstrated that H2A.1 is genetically in the same pathway as the Polδ subunit Pol32 during sister chromatid recombination (SCR) and HR-mediated CAG repeat expansions. Using several genetic assays, we were able to demonstrate that the role of H2A.1 extends beyond repair of repeats and is broadly required for HR-mediated repair (both SCR and BIR) at non-repetitive sequences as well.

Editor comments:My most serious concern involves the antibody used to detect H2A-T126phos. The supplemental documentation (including peptide sequences, etc) in Moore…Krebs 2007 paper is no longer available, and the paper itself seems self-contradictory with regards to which antiserum was used—the Results section seems to imply that the custom antibody was used, but the Materials and methods refers to an Abcam antibody that has been withdrawn from distribution. In addition, Moore…Krebs do show increased signal with their antibody (whatever its source) in response to menadione (2x), phleomycin and MMS (1.5x). Finally, the supplementary figure included as Figure S3 is not well documented and is, in my opinion, of sub-standard quality.

The various antibodies used in the literature are confusing, and we have added more details about our antibody in the Materials and methods to clarify its origin. The antibody used in the Moore…Krebs 2007 paper was a small batch of test antibody made by Abcam. Dr. Krebs has contacted Genetics and requested that they re-post the missing supplemental figure showing its specificity. Abcam then made a larger batch for general distribution, and this batch was not specific and was pulled from the market.

The antibody used in this paper is different from both of these and was a custom antibody made by Aves raised against the phosphopeptide KKSAKA[pT]KASQEL. This antibody showed specificity to the phosphorylated peptide in vitro(Figure 3—figure supplement 2A, now with better labeling). Later, we ordered a second batch of antibody because we were running low, and two different preparations (from 2 chickens) were both not specific – neither are used for any of the experiments in this paper. We have saved back ~70 μl of our more specific Aves antibody for distribution upon request.

I think that, given the dependence the current paper places on this antibody in assigning the T126A phenotypes to a lack of phosphorylation, it is imperative that validation of the antibody be complete and accessibly documented. In particular, because of differences that can exist between peptides in solution and in the context of the full-length protein, it seems to me critical that a phosphatase-treatment control be included, to quantitatively determine what fraction of the signal on Westerns is due to T126phos, and what fraction might be due to cross-reactivity with the unmodified histone (anti-S129phos would serve as a good control in this). I realize that this is not possible with your current protein preps, but it should be pretty straightforward to do the analysis with acid-acetone extracted histones (see Jourquin F., Géli V. (2017) Histone Purification from *Saccharomyces cerevisiae*. In: Guillemette B., Gaudreau L. (eds) Histones. Methods in Molecular Biology, vol 1528 for a protocol).

We did the phosphatase experiment per request, with 3 repetitions and quantification. With both in-solution and on-blot phosphatase treatment the T126ph signal was reduced an average of 30% compared to the level before phosphatase treatment. Thus, the antibody has some preference for the phosphorylated form but also recognizes the unphosphorylated form of H2A.1 in vivo.

On the other hand, the antibody shows 70-80% specificity for H2A.1 compared to H2A.2. Thus, it is reasonably specific for H2A copy 1. This level of specificity is in line with other histone tail antibodies that have been used in the literature. For example, The Abcam H3K27ac antibody (ab4729) is extensively used in the literature (it has 723 references listed on the product information page). By peptide array, this antibody is <30% crossreactive with unmodified H3, <20% crossreactive with K9ac, <20% crossreactive with K36ac, and <20% crossreactive with K18ac. It is still acceptably used to demonstrate H3K27ac.

Therefore, although our H2A.1-T126 antibody is not perfect, we feel it is still useful for detecting differences between the two isoforms of H2A, and we have made conclusions accordingly, with appropriate reference to the background isoform cross-reactivity. On the other hand, we have removed all conclusions relating to specificity to the phosphorylated form, as the 30% preference for the phosphorylated form is too weak to draw definitive conclusions. However, based on our experiments showing a phenotype for the hta1-T126A mutation in vivoin multiple assays, we still speculate in the discussion that phosphorylation of HTA.1-T126 may be important for fidelity of sister chromatid recombinational repair, but include the caveat that this could be due to the polarity of the threonine residue and not necessarily T126ph.

In summary, we have modified our language throughout to only conclude that H2A.1 (copy 1) and a threonine at position 126 is important for our various phenotypes but have left in speculation that this is due to T126 phosphorylation.

I think that it is also important, given the relatively high (and, to my impression, variable) reactivity of the antibody with T126A and S129A mutants (compare Figure S2B with S2D), that quantitative measures be made for all western blots using this antibody. If the antibody cannot be validated, or if substantial and consistent quantitative differences in reactivity with wild-type and mutant proteins cannot be documented, then I think that it will be important to remove references to T126 phosphorylation from the paper, although of course speculation in the discussion is always allowed!

We have now added quantification for all Western blots using the H2A.1-T126 antibody, including duplicate experiments done, so that all experimental values obtained are shown with points and error bars on the graphs. See Figures 3B, 3C, Figure 3—figure supplement 1 and 2B. What can be seen in Figure 3B is that the antibody is specific for H2A.1 (copy one, Hta1 protein). However, there is a background level of reactivity to H2A.2 (averaging 20-34% of WT H2A levels, depending on calculation method, Supplementary file 9). The background reactivity to the mutated T126A H2A.1 tail (averaging 26-30% of WT H2A) is similar to the H2A.2 tail background. Mutation of the H2A1.1 tail at S129 (*hta1-S129A*) also reduces reactivity to a similar 23-24% background level.

Altogether, we conclude the antibody is about 70% specific to the unmutated T126- phosphorylatable version of the H2A.1 tail and has about 30% cross-reactivity to versions of the H2A tail that cannot be phosphorylated on T126.

Statistics. Reviewer 1 (point 4) is being overly gentle. I strongly believe that if an apparent difference is not statistically significant, then one is not justified in calling it a difference. This is a concern that bedevils many of the comparisons, particularly those in Figure 4 and 5. It seems to me that there are two choices here—either a) increase the sample size to the point where there is expected to be sufficient statistical power to distinguish differences, or b) remove mention of these from the text.

This concern would apply to Figure 4B (reductions compared to *hta1-T126A*) and 5B (reductions compared to *hta1*Δ). We have tempered our language in response to your concerns about the statistics; however, we have also included an explanation of Fisher’s Exact Test – that this is a conservative test that can often miss biologically relevant differences. The suppression in expansions is consistent between experiments, but at the current level of difference an excess of 500-1000 colonies/PCR reactions would be required for each mutant to lead to a p-value of <0.05 using Fisher’s. This was not feasible to accomplish and would not have changed the overall conclusions. Given the type of data, we believe it is valid to point out consistent trends.

Genotypes in text and figures. I found myself confused as to which experiments were done with mutant alleles integrated at the endogenous locus, mutant alleles integrated at the endogenous locus but with an hta2-deletion, mutant alleles on a plasmid over a deletion of both HTA-HTB loci, etc. It would really help if the nomenclature in the text and figures made this clear. For example, comparing rad51-del with rad51-del hta1-T126A would appear to me to involve not just a difference in HTA1 genotype, but also a difference in the source of both H2A and H2B (plasmid rather than endogenous locus). It would really help if these differences were transparent to the reader.

To address the confusion pertaining to the genotypes, we have more explicitly stated when point mutations are expressed from a plasmid or integrated at the endogenous locus in the text and/or legends. In terms of the comparisons, we have also clarified this throughout. Per the example listed of rad51-del with rad51-del hta1-T126A, you are correct that the T126A mutant is being expressed from the plasmid. However, these two strains were not compared statistically. We clarified that the comparison is being made from hta1-del to hta1-rad51-del, or hta1-T126A to hta1-T126A rad51del etc. All comparative p-values are listed in Supplementary file 2 and we have added the comparisons to hta1-del and hta1-T126A for the contractions as well.

Other points:“partial suppression of contractions was observed (Table S2)”

Fixed.

“expansions are somewhat suppressed in the hta1-del rad5-del double mutant” is ambiguous; since expansions are at similar levels in both rad5-del strains. Is the difference between hta1-del and hta1-del rad5-del statistically significant?

No, it is not. We have added a clear statement that the difference between hta1-del and hta1-del rad5-del is not statistically significant. However there is a caveat to concluding that template switch has no role that is now also pointed out.

“and were suppressed approximately 3-fold in the hta1-T126A rad57-del mutant”. I think that the difference is 2-fold.

Fixed.

Figure 1C. “wild-type control” for the hta1-del::HTA2 hta2-del strain would be hta2-del; statistical comparisons should be made between these two strains. Don’t think this will change the outcome.

We added the comparison between hta1::HTA2 hta2 and hta2 single to the sentence (p = 7.5 x 10-3).

Figure 2A. These gels/Southerns are not of sufficient quality to determine if there are differences in nucleosome spacing—all that can be inferred it that there is not a major disruption in chromatin structure (i.e. there are still nucleosomes). I think that these gels either should be redone or they should be dropped from the paper.

The quality of these blots is in-line with the expected quality for this type of experiment. We redid the gels/blots several times and got similar quality data as the experiment presented; we do not think it is possible to improve on it. The fuzziness is due to several factors, 1) MNase cleaves between nucleosomes, so exact cutting sites will vary if nucleosome binding sites are not exactly the same in all cells, leading to slightly different-sized fragments, 2) this was a very long gel and so slight differences in fragment size become more obvious, leading to fuzziness, 3) the CAG repeat positions nucleosomes, but the CAG-85 repeat (255 bp) isn’t necessarily centered on a nucleosome – slight shifting between cells/genomes or binding more than one nucleosome is possible.

We have modified our conclusion to “there is no visible, major disruption to the chromatin structure in the *hta1∆* cells compared to the wild-type”, which can be seen from this blot. To complement the less-precise MNase data, we verified that nucleosome protection at the CAG tract is not altered in *hta1Δ* or *hta2Δ* mutants using the more precise Illumina array method, however in this method the MNase digestion is done to completion so only one nucleosome binding site is evident. We think the two types of data complement each other, and that this is an important piece of negative data that eliminates one hypothesis for the difference in CAG instability between *hta1Δ* and *hta2Δ* strains. We have therefore left Figure 2 as-is.

Figure 4A. What do the carat (^) marks mean?

This has been added to the figure legend – suppression from *hta1Δ* single.

Tables S1, S4, S5, S6 and corresponding figures. Underlying data from individual experiments need to be included as a supplementary table (I suggest an Excel file with data from individual experiments). In Figure 5E, some measure of statistical significance is needed. I suggest binning the data into BIR and non-BIR outcomes.

Binning BIR and non-BIR outcomes for 5E is identical to the statistical analysis in 5D, so we refer you to 5D for this information. However, to clarify Figure 5E, we have added more details of how repair type was assessed and the outcomes to the Results section. The raw data for all experiments has been added to the supplementary tables in the Excel document. BIR statistics have been added to the BIR supplementary table; a student’s t-test was used to test significant deviation from the wild-type for each repair outcome.

Reviewer #1:In this study, the authors examine the impact of histone H2A modifications on the stability of a very large CAG/CTG trinucleotide repeat tract, which effectively acts as a fragile site within the DNA. This allows the authors to evaluate the contribution of different repair mechanisms to the stability of the tract (or fragile site) and the impact that H2A-1 modification has on repair pathway selection. The authors demonstrate that the presence or absence of H2A.1 impacts the stability of the long CAG/CTG tract, i.e. there is reduced stability and increased fragility when it is absent. H2A.1 was identified in a screen, H2A.2 does not have the same effect. Given that the difference between the two versions of H2A is a phosphorylatable Thr at position 126 of H2A.1, the authors characterize the importance of this residue in tract stability. Mutation of Thr-126 largely phenocopies the hta1 deletion, with respect to tract expansions. Furthermore, the absence of H2A.1 impacts the repair product profile, apparently suppressing sister chromatid exchange and break-induced replication. Notably, phosphorylation of serine 129, which has previously been implicated in DNA damage response, does not affect tract stability. The role of pol32 deletion suggests a role for efficient D-loop extension in maintaining tract stability.This is a comprehensive and rigorous study that reveals a role for H2A.1 Thr126 ph in stabilizing CAG/CTG repeats. Beyond that, the authors argue convincingly that H2A.1 more broadly influences DNA repair pathway selection to promote higher fidelity repair, at least with respect to tract length, perhaps via sister chromatid exchange and break-induced replication. This analysis makes the work of interest to those working on repeat tract stability, as well as to the broader community interested in genetics and genomics.Major comments:1) The effect of hta1 deletion does not appear to be a result of gross changes in H2A copy number in the cell or altered nucleosome positioning, although subtle differences would be difficult to detect in the assays presented, but could have a significant impact on activity – or pathway selection. Furthermore, there may be effects of cell cycle regulation that are not probed here, nor should they be. But these possibilities should be noted.

We agree that subtle chromatin changes would not be detected, and have modified two sentences to address this possibility. First, in describing the MNase indirect end-labeling result, we have added that there is “no visible, major disruption to the chromatin structure”.

Second, when drawing our conclusion in the paragraph below, we have added “however, subtle differences in chromatin structure in the absence of H2A.1 that are not visible by these assays may still have an impact on repair pathway selection.”

Previous work has shown that gene dosage of *HTA2-HTB2* can amplfy to form a minichromosome when *HTA1-HTB1* is absent (Libuda and Winston, 2006). We tested whether gene amplification of *HTA1* or *HTA2* had occurred in any of our strains and found no instances of gene amplification at the genomic level in the strains used in this paper (See new Supplementary file 8).

For cell cycle effects, we have added an experiment to show that hta1 deletion cells have a normal DNA damage checkpoint activation and resolution (by Rad53 phosphorylation). We have noted the possibility that another aspect of cell cycle regulation could be affected.

2) Unlike H2A-S129, the authors demonstrate that H2A-T126ph is not induced by the DNA damaging agents MMS and HU. As a result, the authors suggest that this modified form of H2A is normally present in the chromatin. Is there any evidence for preferential localization of H2A- T126ph at the tracts tested in this study or is it distributed across the genome?

We performed a ChIP experiment to test the association of H2A.1-T126ph (as detected by our antibody) with the CAG repeat, and detected a specific association at 40 min into S phase.

Previous ChIP at this same CAG location showed association of Mre11 at 20 min and γH2AX association also at 20 min and peaking at 40 min., then disappearing. Therefore, our data are consistent with H2A.1-T126ph being associated with replication-associated DNA damage at the repeat.

The antibody we used was purified from chicken eggs, and was quite dilute. Attempts to concentrate it resulted in a significant loss. We found we needed to use about 20 μl per sample for the ChIP to be successful, which amounted to 80-100 μl per cell cycle experiment. As noted in the response to the Editor’s comments, we were running low on antibody, and ordered more, but both new preparations were not specific. Therefore, we are unable to perform any more ChIP experiments, as we want to save back some antibody for requests from colleagues (the dilute preparation works fine for Western blots). We would love to do a genome-wide experiment, but it is not possible at this time.

3) The focus throughout the manuscript is on tract expansions, although contractions were also examined. Notably, while hta1 deletion influences both expansions and contraction, some of the other genetic backgrounds reveal differences in the impact on expansions versus contractions. Can this tell us anything about the mechanisms that result in tract instability – expansion versus contraction? Why does knocking out NHEJ in (lif1 deletion) or PRR (rad5 deletion) in combination with hta1 deletion lead to increased expansions with no detectable change in contraction frequency?

Our lab and others have examined the effect of various repair pathways on CAG repeat expansions and contractions, and there is an extensive literature on this topic. To simplify, gap- filling type repair or HR/PRR involving DNA synthesis tends to lead to expansions (due to slippage or improper flap processing), whereas DSB repair through NHEJ or SSA often leads to contractions. Our data support a model in which H2A.1 is facilitating a step of HR, like D-loop extension, to occur with fidelity. Since this is a synthesis step, expansions can result if it occurs inappropriately (as in the hta1 deletion or hta1-T126A mutant). Contractions could occur during this same process if the template strand forms a structure, or during repair of occasional breaks.

To address your specific question, knocking out PRR (rad5 deletion) on its own mildly increases instability (both expansions p=0.02 and contractions p=0.03). There is no further increase in combination with hta1 delete (hta1rad5), in fact the two mutations are less than additive and are similar to the rad5 levels, suggesting that Rad5 may work upstream of Hta1. We have now mentioned this in the presentation of Figure 4A and 4B.

Deleting lif1 in the hta1 background does suppress the elevated contractions observed in the hta1 background, suggesting that those contractions are occurring through NHEJ of a DSB. This result suggests that the somewhat increased breaks that occur within the CAG repeat in the hta1 background can be healed by NHEJ to cause contractions. We have not focused on this result, as it fits already known and published mechanisms of contractions and is likely not reflecting a specific function of Hta1 in the cell (note that neither fragility nor contractions were increased in the hta1-T126A mutant, in agreement with this interpretation; this data is now included in Figure 1—figure supplement 1B and Supplementary file 2).

We focused on the expansion phenotype in the absence of H2A.1 since it is expansions that are increased in the T126A mutant. We have added the following sentence to the manuscript to more overtly state this:

“Although contraction frequency is increased in the hta1Δ mutant, the frequency is not significantly increased from wild-type in the hta1-T126A or hta1Δ::HTA2 mutant; therefore we conclude that the H2A.1 sequence-specific role is in preventing expansions.”

4) Similarly, the hta126 point mutations (T126A and T126E) both increase expansions but contractions are unaffected, unlike the hta1 deletion, which affects both expansions and contractions. The authors suggest that it is dynamic phosphorylation of this threonine residue that is important. Is this a result of needing dynamic phosphorylation within a single histone core or simply the need to have pools of phosphorylated and unphosphorylated H2A.1 in the cell? Is either the hta1-T126A or hta1-T126E dominant negative? What is the effect on tract stability (expansions and contractions) of co-expressing both mutations?

This question of whether dynamic phosphorylation is occurring on a single histone core is an interesting one, but very difficult to address with the tools at hand. We thought about co- expressing both mutations (T126A, T126E) but since they are the same residue we would either have to introduce 2 copies of H2A.1 in a haploid cell (an unnatural situation with unknown consequences) or do the experiment in a diploid cell, necessitating redoing all the controls and single mutants in this situation. Also, since each single (T126A and T126E) gave a similar phenotype (Figure 3A), it might be hard to distinguish an effect of co-expression. However, we did test the effect of a T126A S129A double mutant on CAG expansions, which did not show any synergism or suppression (now in Figure 3A). The hta1-T126A point mutation does not appear to be dominant negative since the presence or absence of the Hta2 protein did not affect expansions (Supplementary file 2) or DNA damage sensitivity (Figure 1-figure supplement 2).

5) How does the suppression of unequal sister chromatid recombination in the absence of hta1 correlate with a decrease in higher fidelity repair at these fragile sites in the absence of hta1? That is, suppression of unequal SCR would seem to indicate that there is less lower fidelity repair. Would a decrease in misaligned recombination not be expected to stabilize the tracts? I understand that the assay is meant to measure spontaneous recombination between sister chromatids, but it does rely on misalignment.

It is true that one type of infidelity could be the misalignment required for an SCR readout in this assay. A second type of infidelity could occur during the SCR process itself after it initiates, for example during D-loop synthesis or resolution. Since we observe a decrease in SCR in the hta1 mutant but an increase in CAG expansions that are HR dependent, we conclude that the defect in efficiency of SCR must occur after the initiation step, such as during the D-loop synthesis (as a decrease in misaligned recombination would be expected to stabilize the tracts, as you point out). We have now added a sentence to this effect. Our data showing that BIR, a process with extensive D-loop synthesis, is also decreased in the hta1 mutant supports this interpretation.

To further address this point, we have added an SSA assay (Figure 4E), and find no defect in an hta1 mutant. This supports that H2A.1 is not affecting the alignment step of recombination, and this point is now made in the paragraph presenting the SSA data.

Minor points:1) There seems to be some language missing in the very first sentence of the Abstract.

Added the three missing words – Repetitive and structured.

2) In the fourth paragraph of the Introduction, it would be useful to refer to the tracts as CAG/CTG or CAG:CTG tracts, or some variation of this, because the nucleosome positioning cited was specifically in CTG repeats (which obviously have CAG on the opposite strand).

This has been fixed throughout the Introduction.

3) Page 5 – 4th line from the bottom of the second to last paragraph – it seems that this refers Supplementary file 2.

Fixed.

4) It might be worth discussing the choice of statistical tests and how conservative they might be, particularly given the discussion in a couple of places of results that are interesting but not statistically significant, e.g Figure 4B,D, Figure 5B.

We have added a few sentences describing that Fisher’s is a conservative test and have more explicitly pointed out when trends are interesting but do not reach the level of statistical significance with Fisher’s.

5) There is some discussion of Western blots used to explore phosphorylation within H2A.1 (second paragraph from the bottom). The authors suggest that the reduced signal is a result of reduced accessibility of the antibody, whereas the more straightforward interpretation is that there is reduced phosphorylation, which is the interpretation discussed in the Discussion.

We have now mentioned both possibilities in both the Results and Discussion (this is in regards to reduced antibody recognition of the hta1-T126A and hta1-S129A mutants). We favor that the decreased signal is due to antibody recognition rather than T126ph being dependent on S129ph for three reasons:

1) CAG expansion frequency is elevated in the T126A mutant but not the S129A mutant. If T126ph depended on S129ph, then we would expect the hta1-T126A and hta1-S129A mutants to show equivalent CAG expansion phenotypes. As this is not the case it is unlikely that T126 requires phosphorylation at the S129 residue.

2) S129ph is induced by MMS but T126ph is not. If T126ph was dependent on S129ph, it should also depend on induction of MMS damage.

3) The H2A.1T126ph antibody was raised against a peptide that contained an unmodified serine at position 129. Although the antibody can recognize T126ph in the presence of S129ph (+MMS, Figure 3C), it appears that alanine at position 129 impedes antibody recognition. The signal in the S129A mutant is equivalent to both the T126A and hta1-del mutants, in-line with background reactivity of the antibody.

These interpretations have been expanded upon and clarified in the Results section. We further expand on the interpretation in the Discussion.

Reviewer #2:In this manuscript, the authors present compelling evidence for a difference between the two different histone H2A proteins in *S. cerevisiae* with respect to the stability of CAG repeats. A deletion of HTA1 confers instability while a deletion of HTA2 does not.Furthermore, they provide additional evidence that this is not due to a change in histone levels but rather to the T126 present in HTA1 that is not present in HTA2. Finally, the provide evidence, confirming previous studies, that T126 is phosphorylated. Additional studies examine the instability phenotype when combined with other mutations that affect stability. Overall the results are interesting as they provide the first new information about a qualitative difference between these two H2A proteins since a paper by Norris and Osley in 1987.1) The main issue with these studies, which is difficult to address, is whether it is the phosphorylation of T126, or some other aspect of T126 that is responsible for the phenotype. There are no results that distinguish between these possibilities, although the latter is not acknowledged and it should be. Both T126A and T126E changes both cause the phenotype, which the authors interpret as a requirement for dynamic phosphorylation, but which could also be interpreted as loss of the T is responsible for the phenotype. In addition, the authors are unable to detect a change in the level of T126 phosphorylation in response to several DNA damaging agents and in several mutant backgrounds. It might be informative to try a different polar amino acid, such as S. If that had a wild-type phenotype, that would argue against the idea that phosphorylation is important.

We concede that some other property of the threonine at that position in the H2A C-terminal tail could contribute to DNA repair and have now explicitly stated this possibility: “Although our results implicate a requirement for dynamic modification of H2A.1-T126 in DNA repair, some other physical property of threonine at position 126 may be important for repair, rather than phosphosphorylation *per se*.” see Discussion.

Our Western quantifications show that the antibody recognizes H2A.1 over H2A.2 with ~70% specificity, however the new phosphatase experiment showed that the specificity for the phosphorylated form in vivois lower, only 30%. Therefore, we altered our language to be more conservative with regards to conclusions related to phosphorylation of T126. Nonetheless, the presence of H2A.1 at the CAG tract by ChIP at a specific time during S phase coincident with S129ph, does support that it could be the phosphorylated form that is important for the observed phenotypes of the hta1-T126A mutant. Serine would be the closest polar amino acid to substitute for threonine, but we were worried about the interpretation of a T126 to serine mutation, as serine can also be phosphorylated.

2) Much more information should be provided about the screen. What strains were screened – was it something broad like the deletion set, or was it more focused? Please provide details in Results and/or Materials and methods.

This screen was originally published in Gellon et al., 2011 and *hta1Δ* is listed as a screen positive in the supplement of that paper. We have added that the Stanford deletion set was screened, but reference Gellon 2011 in lieu of adding the methods and results from the screen.

3) Figure S1 – This figure should be included in the main figures in the text as it provides information about the screen and it shows the initial data about the nature of the histone mutations that increase the rate of 5FOA resistance.

The fragility figure has been added to Figure 1.

4) The TRT nomenclature has not been used for many years. I’m don’t see the point of using it here as it might be confusing to those who are not familiar with it. Please just use HTA1-HTB1 and HTA2-HTB2.

This has been changed.

5) The authors point out that there is an amplification of HTA2-HTB2 when there’s a deletion of HTA1-HTB1. Therefore, in the strains used in this paper that contain an hta1 deletion, what has happened to HTA2? Has this duplication occurred? Also, they write that “the H2A.2 gene is not upregulated…” but if this amplification occurs, doesn’t that upregulate expression? In addition, they should use HTA2, not H2A.2 when referring to the gene.

We have clarified that “upregulation” refers to transcriptional upregulation in the text. We have now tested by qPCR for HTA gene levels in the *hta1*Δ and *hta2*Δ mutants and found no evidence of gene duplication of HTA2 in any of our *hta1*Δ mutant strains. These results have been added to supplemental Supplementary file 8.

6) Figure 3 – In the S129A mutant, the level of signal for T126ph is reduced to background – that seen in the T126A mutant. The authors interpret this to mean that the S129A mutant binds the anti-T126ph antisera less well. However, it’s also possible that S129 is required for phosphorylation of T126. The authors should include this possibility.

We have noted this possibility in the Results section. However, we favor that antibody recognition is responsible for the decreased signal for three reasons:

1) CAG expansion frequency is elevated in the T126A mutant but not the S129A mutant. If T126ph depended on S129ph, then we would expect the hta1-T126A and hta1-S129A mutants to show equivalent CAG expansion phenotypes. As this is not the case it is unlikely that T126 requires phosphorylation at the S129 residue.

2) S129ph is induced by MMS but T126ph is not. If T126ph was dependent on S129ph, it should also depend on induction of MMS damage.

3) The H2A.1T126ph antibody was raised against a peptide that contained an unmodified serine at position 129. Although the antibody can recognize T126ph in the presence of S129ph (+MMS, Figure 3C), it appears that alanine at position 129 impedes antibody recognition. The signal in the S129A mutant is equivalent to both the T126A and hta1-del mutants, in-line with background reactivity of the antibody.

These interpretations have been expanded upon and clarified in the Results section. We further expand on the interpretation in the Discussion. We agree that T126ph could be dependent on the presence of a serine 3 amino acids away (but not its phosphorylation), explaining the H2A.1-specific phenotypes, and have suggested this possibility in the Discussion.

7) The authors justification for testing a rad5 mutant is to test if chromatin modifications might make a difference in the hta1 mutant phenotype. This requires change or more justification – how does rad5 relate to chromatin modifications?

Our previous results (House et al., 2014) demonstrated that Rad5-mediated repair at CAG repeats was influenced by chromatin modifications (H4K16ac), which was our original justification/interest in performing those experiments. However, given the reviewer’s concern, we have changed the justification to “Rad5-dependent post-replication repair has previously been shown to be a source of expansions during low-fidelity repair at CAG repeats.”

8) Have the authors tested the level of Pol32 protein in the various hta1 and htb1 mutants to see if it is altered? That could account for some of the phenotypes.

We tested POL32 mRNA levels by RT-qPCR in both WT and hta1-del strains and found no difference (see Supplementary file 10).

Reviewer #3:The manuscript by House et al. reports a role for H2A.1 threonine 126 phosphorylation in triplet repeat maintenance in yeast. Using mainly genetic approaches, H2A.1 is shown to be required for CAG stability while the other copy of H2A, H2A.2, is dispensable.Mutation of Thr126 in H2A.1, which is not found in H2A.2, renders yeast cells CAG unstable. The instability of CAG repeats was further shown to require HR and Pol32, which appeared to function through sister chromatid recombination. BIR was also shown to be defective in H2A.1 threonine 126 mutant cells. Overall, this is a well-executed study that reports an interesting differential involvement of H2A genes in yeast that appears to act through the phosphorylation of H2A.1 Thr126. This work is interesting from several angles including a new histone mark involved in CAG repeat stability as well as the suggesting that yeast may contain histone H2A variants within the H2A1 and H2A2 that are normally considered to be core H2A histones. While this work will be of broad interest to researchers in several biological sciences, which make this work a strong candidate for publication, some additional work should be performed to address the following questions.Main questions.1) The major issue with this work is that there is no mechanism for how T126p promotes CAG stability. While the publication of this work should not rely on this, some additional experiments/discussion should be added to better place this modification within this pathway. For instance, is this a Mec1/Tel1-dependent event and if so, are these kinases required for CAG stability? Does this modification occur at CAG sites? This is a very important question to support the model as any histone modification could act indirectly. To place this modification at the CAG repeats would go a long way to support the model proposed in Figure 6A. Along the same line of enquiry, are the levels of H2A.1 and H2A.2 the same at CAG repeats or are their differential loading of these histones which would also promote the CAG stability by having phosphorylatable H2A histones in proximity to these repeats.

We performed the requested ChIP experiment and were able to demonstrate that H2A.1 is specifically enriched at the CAG repeat 40 minutes after α-factor release, which is the time at which S129ph at the repeat is maximal (House et al., 2014). This result suggests that H2A.1 is specifically incorporated at replication-associated damage at the expanded repeat tract. Since we do not have an antibody specific to H2A.2, we could not track level of this histone variant at the CAG tract, but we did confirm that H2B and H3 levels are unchanged (Figure 3D, 3E).

We tested the effect of multiple kinases known to phosphorylate in response to DNA damage including Mec1, Tel1, Rad53 (and Rad9), Chk1 and Dun1. None of them showed a significant reduction in H2A.1-T126 antibody signal (Figure 3—figure supplement 1E). However, our new phosphatase experiment data indicate that our antibody shows only a mild preference for the phosphorylated form of the protein, therefore this approach to finding the relevant kinase is not ideal. Since there are over 100 kinases in the yeast genome, identification of the kinase will require a different approach, beyond the scope of this study.

2) The potential crosstalk between S129 and T126 is intriguing. While the Results section proposes that the reduction in T126p by mutation of S129 is perhaps an artifact of the antibody, the discussion suggests that this may be a real result. This is somewhat confusing for the reader. Some additional work should be performed to address this question. Either antibodies can be used to check the specificity of the T126p epitopes or genetic analyses could be performed to see if there are interactions between the loss of T126 and S129. While the genetics seem to support a separation of function between these two modification sites for CAG repeat stability, additional work would help strengthen this idea which is an important one and an issue that is currently unresolved in this manuscript.

The genetics don’t support that T126ph is dependent on S129ph, as we saw no increase in expansions for a S129A mutant. Also, we have now tested CAG expansion frequencies in the S129A T126A double mutant, and see no increase over the T126A level. Therefore, the genetic data do not support a dependence of T126ph on S129ph. We now conclude more clearly that the decrease in antibody recognition due to the S129A mutation is likely due to antibody epitope recognition, rather than a co-dependency of the two modifications. This can be found in the Results section and expanded upon in the Discussion.

3) It would be interesting to perform DNA damage sensitivity assays for H2A.1, H2A.2 and the phospho site mutants to further demonstrate the importance of these different H2A genes and these sites in DNA damage response pathways.

These assays have been added to the supplement – we did not see significant sensitivity of the *hta1Δ* mutant to the agents tested, although it is perhaps mildly sensitive to camptothecin and phleomycin.

4) Figure 3B should be ran on the same gel. Levels of phosphorylation between different samples are difficult to compare when ran on different gels. Given that these are only 6 samples, there should not be any issues in performing this experiment.

The samples were run on the same gel, but the bar was used to delineate the different treatments. We have removed the bar to address the reviewer’s concern (now Figure 3C). We have also added quantification of this and duplicate experiments.

5) The data in Figure 5E is interesting. For the hta1 mutant, the repair types are only slightly reduced for BIR, especially compared to pol32 mutants. However, this mutant seems to be the only one where GCs are scored. This seems interesting and deserving of a discussion to explain why this may be. Although not required, it would have been nice to have the T126A mutant analyzed here to ensure that all of these phenotypes for hta1 are occurring through this phospho-site.

GCs were scored in each mutant, however the values are small enough to not be clearly visible in the WT. For *pol32Δ*, this was a mistake which has now been corrected: GCs (maybe due to aborted BIR) are also increased in this mutant (to 9.8%, Supplementary file 7). We think that the increase in alternative repair types in both the *hta1Δ* and *pol32Δ* mutants is indicative of the BIR defect, which we have now clarified in the text. It does seem that the proportion of GC events compared to NHEJ is higher for *hta1Δ* compared to *pol32Δ* which could indicate that D-loop synthesis is relatively less impaired in *hta1Δ*, but we were hesitant to make a point about this since the reason is speculative. We also tested BIR in the *hta1-T126A* mutant and found no significant defect. All raw data for the BIR assays can now be found in Supplementary file 7.

Minor questions.1) The authors should consider editing the first sentence of the Abstract, “DNA are sites of genomic instability”. This sentence distracted this reader from the beginning.

The first three missing words have been replaced – Repetitive and structured.

2) In the Introduction, second paragraph, H2A/H2AX K15 Ub by RNF168 which is bound by 53BP1, as well as TIP60-mediated H2A/H2AX K15 acetylation should be added to the discussion of modifications on H2A histones.

As originally written, this was listed as modifications that have been directly shown to localize to sites of DNA damage, either by ChIP, foci IF, or microirradition. To include K15ub/ac as requested, we broadened the language from “detectable at breaks” to “contribute to DNA repair.”

3) The model in Figure 6B is confusing. It is labelled No H2A.1 but the figure shows this histone without the phosphorylation. I think the authors would like to say “No H2A.1 T126 phosphorylation”. This should be edited for clarity.

We have changed the labeling on the model figure to emphasize the specific role for H2A.1, and changed the color of the histones and labeling in part B to indicate the situation where only H2A.2 (not phosphorylated in our model) is present.

[Editors’ note: the author responses to the re-review process follows.]

The major concern remains that of the antiserum used. It is clear now, from the work now done in characterizing the antiserum, that this is a reagent with complex specificity; in addition to the substantial reactivity remaining after phosphatase treatment, specific reactivity appears to require both T126 and S129, with an indication that phosphorylation of S129 may reduce reactivity. Given the complex nature of the reactivity of this antiserum, it seems clear that experiments using it cannot be interpreted unambiguously.

The antibody shows on average 70% specificity for H2A.1 compared to H2A.2. Thus, it is reasonably specific for H2A copy 1. This level of specificity is in line with other histone tail antibodies that have been used in the literature. For example, The Abcam H3K27ac antibody (ab4729) is extensively used in the literature (it has 723 references listed on the product information page). By peptide array, this antibody is <30% cross-reactive with unmodified H3, <20% cross-reactive with K9ac, <20% cross-reactive with K36ac, and <20% cross-reactive with K18ac. It is still acceptably used to demonstrate H3K27ac.

Therefore, although our H2A.1-T126 antibody is not perfect, we feel it is still useful for detecting differences between the two isoforms of H2A, and we have made conclusions accordingly, with appropriate reference to the background isoform cross-reactivity.

Regarding the “complex specificity”, the antibody was raised against the T126ph form of the H2A.1 tail, so it is natural and expected that it would recognize the amino acid sequence of the tail (e.g. threonine at position 126 and serine at position 129). The fact that the antibody recognizes the sequence it was raised against best (T126ph and S129 not ph) is also not surprising, and we can interpret experiments using it appropriately since we did all the proper control experiments to characterize the reactivity under each condition. Nonetheless, in this version, we have focused on its ability to distinguish between the two H2A isoforms, rather than the state of T126 or S129 phosphorylation.

The antibody is less specific for the phosphorylated form of the H2A.1 tail. Although it shows high specificity in vitro, it has only a 30% preference for the phosphorylated form according to our in vivophosphatase experiments. Therefore, we have removed all conclusions relating to specificity to the phosphorylated form and either moved that data to the supplement or qualified our conclusions based on the recognition of both phosphorylated and non-phosphorylated forms. However, based on our experiments showing a phenotype for the hta1-T126A mutation in vivoin multiple assays, and previous published results that it is phosphorylated in vivo, we still speculate in the discussion that phosphorylation of HTA.1-T126 may be important for fidelity of sister chromatid recombinational repair, but include the caveat that this could be due to the polarity of the threonine residue and not necessarily T126ph. In summary, we have modified our language throughout to only conclude that H2A.1 (copy 1) and a threonine at position 126 is important for our various phenotypes but have left in speculation that this effect could be due to T126 phosphorylation.

Given that these experiments remain a major fraction of the manuscript, I do not think that it would be appropriate to proceed further with the current submission.

The experiments using the antibody are not a major fraction of the paper: they are 3 experiments that constitute part of 1 figure out of 5 figures (3 experimental panels out of 17). Specifically, in this revision, we have focused on the experiments using the antibody to distinguish between the two H2A forms, H2A.1 and H2A.2: Figure 3B shows the specificity for H2A.1. Figure 3 C and D shows that H2A.1 specifically locates to the CAG tract at the same time as H2A-S129ph. Both reviewer 2 and 3 commented that this was an important addition to the paper “The new data showing that H2A.1 is enriched at triplet repeats is an important addition, further demonstrating a direct role for this histone at repeat sequences.” (reviewer 3). Figure 3E shows that the T126A phenotypes are not due to an indirect effect on S129 phosphorylation, which is an important control. These three experiments are focused on the H2A.1 and hta1-T126A phenotypes and therefore support the main points of the paper. We also note that all of our genetic and other data in Figures 1, 2, 3A, 4, and 5 do not depend on this data, so even if it were discounted the story is strong and stands on its own. We disagree that the data obtained using the H2A.1-T126 antibody is a major fraction of the manuscript, but nonetheless, as described above, argue that it adds important information that complements our genetic data.

The reviews are copied below and we hope that they may help you should you decide to revise the manuscript for submission elsewhere.We are sorry that we cannot be more positive on this occasion, but hope that you appreciate the reasons for this decision.

Comments to the Authors:

Reviewer #1:This revised manuscript effectively deals with most of the earlier concerns. The apparent role of H2A.1 in D-loop-mediated recombination is very interesting. However, the issue of T126 phosphorylation hasn't been adequately addressed and this limits the mechanistic understanding of the data.The results for H2A.1 phosphorylation using the T126 antibody are confusing as presented. Basically, the authors have an antibody specific to H2A.1, with some preference for a phosphorylated protein, but not enough to make any real conclusions about the role of T126 (phosphorylation vs. structural). It seems, therefore, that this section should be reduced and re-framed in that context. Perhaps the order of the results in the new section dealing with the antibody specificity could be re-arranged. Perhaps going through the specificity of the antibody for H2A.1 versus H2A.2 then the localization of H2A.1 at the CAG repeat, followed by looking at phosphospecificity (or lack thereof), which limits the interpretation of the role of T126.

We have reframed and restructured this section as suggested by the reviewer, first presenting the specificity to H2A.1 (now Figure 3B) and experiments that show H2A.1-specific phenotypes (regardless of phosphorylation state) (Figures 3C, 3D, 3E). We are upfront about the fact that the antibody was originally raised against the phT126 residue with the aim of testing that modification in our system, but then detail our testing that showed the specificity is greater for the H2A.1 C-tail versus the H2A.2 tail, though there is also a recognition of the T126 phosphorylated form of the H2A.1 tail. (Figure 3—figure supplement 1).

Accordingly, we have re-named the antibody H2A.1-T126 (instead of H2A.1-T126ph) and re-labeled the relevant figures.

Can you not take your histone preps and perform a mass spectrometry analysis to look at H2A.1 phosphorylation in different genetic contexts?

We considered this experiment but believe it will not be successful for the following reasons. There are 4 published papers that investigated the yeast phospho-proteome using mass spectrometry (Smolka et al., 2007; Chen SH et al., 2010, Bastos de Oliveira et al., 2015, and Zhou et al., 2016). The paper from the Elledge lab was particularly comprehensive, using SILAC technology coupled with mass spec identification and testing not only MMS, but HU in S phase cells, and irradiation of mitotically arrested cells. None of the papers detected H2A.1 T126 phosphorylation, though it was detected by 2D protein gels of labeled histones in Wyatt et al., 2003. I communicated with Steve Elledge about potential reasons for this. Because T126 is surrounded by 2 lysines (K124, K127), it will end up on a peptide only 3 amino acids long after trypsin digest, which will not be detected by mass spec (6 residues is the typical minimum peptide detected). We now mention this issue in the text. Steve Elledge said he didn’t think that this short peptide would be detected by SILAC labeling either. Because we don’t have another good way to verify T126ph besides what has already been published and our imperfect antibody, we have focused the revised manuscript on the different H2A subtypes.

I don't see how, as stated by the authors in the discussion, one can determine whether T126ph is dependent on S129ph.

We have clarified our language on this point. Our conclusion is that since the hta1-S129A mutant has no CAG expansion phenotype, and S129ph occurs unimpeded after MMS treatment (DNA damage) in the T126A mutant, the phenotype in the hta1-T126A mutant is not due to an indirect effect on the ability of S129 to be phosphorylated. Figure 3E is included to make this point.

Throughout the discussion there are places where T126ph is assumed to be the relevant form of the protein, including the very last sentence.

We have now modified the discussion to be more even-handed between the following possibilities:

1) T126 may be locally phosphorylated at the CAG repeat during repair and required for repair factor recruitment; 2) The amino acid sequence in the H2A C-terminal tail is required for DNA repair protein recognition (rather than phosphorylation of the residue, the reside itself is required), or 3) modification of T126 affects modification of other tail residues, such as S122ph, K124ac, or K127ac that are then required for efficient repair. We have also modified the last sentence to remove the word ‘phosphorylation’.

The authors argue that the reduced signal with S129A is a result of reduced recognition because of the mutation. Is this not also the case with T126A, then (e.g Figure 3B)?

Yes, that is the case. We conclude that the antibody recognizes the sequence of amino acids in the H2A.1 C-terminal tail, including T126 and S129. Therefore, upon mutation of the T126 residue, the antibody recognition is significantly diminished. As stated, “The signal is significantly diminished in the *hta1Δ* and *hta1-T126A* strains to 20-30% of WT levels, but not in the *hta2Δ* mutant (Figure 3B. Supplementary file 9). We conclude that the antibody is specifically recognizing the H2A.1 protein isoform containing threonine at position 126 in the tail, with a low level of background reactivity to the H2A.2 isoform.” And “we conclude that the *hta1-S129A* mutation likely disrupts the antibody epitope rather than affecting H2A.1-T126 modification.”

Throughout, some more care is required in describing assays and conditions used for the more general reader. For example, why is the MMS treatment relevant in testing the phospho-specificity of the T126 antibody?

We have added additional explanation to the text describing assays and conditions. MMS is used in the phosphatase experiment in Figure 3—figure supplement 1B to induce S129ph as a positive control, now stated in that legend. It was used in Figure 1-figure supplement 2 spot assays to induce DNA damage, now stated in that legend. It was used in Figure 1-figure supplement and 2 Westerns as a DNA damaging agent to induce S129ph and test whether H2A.1 expression or T126ph was similarly induced, now stated “To determine if H2A.1 expression is damage inducible, H2A.1 levels were monitored after exposure to MMS, a DNA base alkylating agent that causes abasic sites that can be converted into single and double strand breaks.” An interpretation of the results was added a couple sentences later. Better explanations of both the SCR and SSA assays have also been added (see details below).

The new SSA assay cartoon figure is poorly labeled so that the intermediate formed is not clear. How does misalignment lead to gene conversion in the SCR assay?

The SSA figure has been updated to be more clear and more details were added to the figure and legend.

For SCR assay clarification, we have added the sentence (p9), “In this assay, misaligned recombination between two *ade2* null alleles can result in gene conversion to a functional *ADE2* allele and the strain is converted from Trp^+^Ade^-^ to Trp^+^Ade^+^” and directed the reader to the Mozlin, Fung and Symington, 2008 publication that extensively details this assay for more details. We have also added “X’s” to the diagram to indicate the illustrated cross-over that would yield Ade2+ colonies, and more fully explained the events shown in the legend.

Reviewer #2:The revised manuscript has been extensively revised and is considerably stronger than the first version. The authors have done an excellent job of responding to all of the comments. I have just two very minor writing comments, listed below.1) Since genes are deleted, not proteins, the sentence should read, “…deletion of the gene encoding histone H2A.1…” The same correction should be made a few lines below. And later in the manuscript for the lif1 experiment.

These corrections have been made. Changed to: “Deletion of the genes encoding the two copies of H2A differentially affect CAG repeat stability…” The other sentences were also corrected.

2) Change “…hybridization of the antibody…” to “…recognition by the antibody…” or something similar.

Changed to: Although this antibody was specific to H2A-T126ph by peptide dot blot (Figure 3—figure supplement 1B) phosphatase treatment of cell extracts only resulted in a 30% reduction in antibody recognition (Figure 3—figure supplement 1C, Supplementary file 9)

Reviewer #3:The revised manuscript by House et al. has provided substantial new data to support a role for H2A.1 in triplet stability and various HR pathways. Although a direct role for H2A.1 T126 phosphorylation has been dampened due to issues with the specificity of the antibody, this work still reports some very interesting findings about the differential role of the two H2A genes in yeast, suggesting that yeast in fact may have an H2AX like variant and a canonical H2A. The new data showing that H2A.1 is enriched at triplet repeats is an important addition, further demonstrating a direct role for this histone at repeat sequences. It is unclear why H2A.2 was not similarly ChIPed, perhaps in a H2A.1 mutant, but given the control to show that H3 and H2B levels are constant at these regions, it is reasonable to assume that this is specific for H2A.1. The additional data showing that BIR levels are normal in the H2A.1 T126 mutant compared to H2A.1 delete is also a nice addition, as it suggests additional residues on H2A.1, independent of T126, that are involved in BIR. Although there remains several unanswered questions here, overall, this study is now sufficiently revised to support publication as the authors have addressed all of this reviewer’s previous concerns.Minor issues.1) Mammalian H2AX T126 mutants have been studied previously. In Xie et al., 2010 (PMID: 20703100), no sensitivity to IR or defects in HR were observed in H2AX KO ES cells reconstituted with H2AX T126V. It would have been perhaps appropriate to discuss this study in the conclusion for the speculation of translating these results with the yeast protein to mammalian H2AX.

A reference to this result and the citation was added and discussed w.r.t. our results in the last two sentences: “A T136V mutation did not decrease survival after IR or DSB-induced HR in mouse cells, but gap repair and repair fidelity were not analyzed (63). Thus, it would be interesting to test whether mammalian H2AX T136 plays a role in repair fidelity analogous to the role described here for yeast H2A.1-T126.”

Given the lack of phenotype in the BIR assay for H2A.1 T126, the model (in Figure 6A) is still confusing since doesn’t this show that T126 is dispensable for BIR, which the model doesn’t suggest.

For the purposes of the model figure, we had to decide whether to show T126 as phosphorylated or not. We chose to show it as we favor a model where phosphorylation has a role, even though it may not be relevant for all steps, but added this explanation in the legend “Though T126ph is shown as present during BIR/BFR for continuity with the previous diagram, our data suggest that the T126 identity has minimal effect on this step in the context of BIR (BFR was not tested).”

When discussing the step during recombination at which H2A.1 is working, we added this explanation of why T126 may be dispensable for BIR: “A second possibility is that H2A.1-T126 is required to promote a later step in the process such as D-loop resolution or re-establishment of the chromatin structure of the repaired gap. This could explain why the H2A.1-T126A mutant did not have a discernable effect on BIR, which does not include re-engagement of the extended D-loop with the initiating DNA molecule.” To better illustrate this point in the model, we removed the T126ph residue from the template strands.

We also added another possible explanation of the effect of hta1 deletion but not hta1-T126A on BIR in the description of the model: “We note that CAG fragility, CAG contractions, and BIR that all involve an initiating DSB were dependent on H2A.1 but not the T126 residue. Therefore, our data are consistent with H2A.1 levels affecting DSB repair while the H2A.1 sequence-specific role is during gap repair/SCR.”

Together, these changes clarify our model and should eliminate the confusion.